# Printable microscale interfaces for long-term peripheral nerve mapping and precision control

Timothy M. Otchy 1,2,3✉, Christos Michas 4, Blaire Lee1, Krithi Gopalan4, Vidisha Nerurkar1, Jeremy Gleick4, Dawit Semu1, Louis Darkwa1, Bradley J. Holinski5, Daniel J. Chew5, Alice E. White 4,6 & Timothy J. Gardner 1,2,3,4,7✉

The nascent field of bioelectronic medicine seeks to decode and modulate peripheral nervous system signals to obtain therapeutic control of targeted end organs and effectors. Current approaches rely heavily on electrode-based devices, but size scalability, material and microfabrication challenges, limited surgical accessibility, and the biomechanically dynamic implantation environment are significant impediments to developing and deploying peripheral interfacing technologies. Here, we present a microscale implantable device – the nanoclip – for chronic interfacing with fine peripheral nerves in small animal models that begins to meet these constraints. We demonstrate the capability to make stable, high signal-to-noise ratio recordings of behaviorally-linked nerve activity over multi-week timescales. In addition, we show that multi-channel, current-steering-based stimulation within the confines of the small device can achieve multi-dimensional control of a small nerve. These results highlight the potential of new microscale design and fabrication techniques for realizing viable devices for long-term peripheral interfacing.

[1] Department of Biology, Boston University, Boston, MA 02215, USA. [2] Neurophotonics Center, Boston University, Boston, MA 02215, USA. [3] Center for Systems Neuroscience, Boston University, Boston, MA 02215, USA. [4] Department of Biomedical Engineering, Boston University, Boston, MA 02215, USA. [5] Bioelectronics Division, GlaxoSmithKline, Stevenage, Hertfordshire SG1 2NY, UK. [6] Department of Mechanical Engineering, Boston University, Boston, MA 02215, USA. [7] Present address: Knight Campus, University of Oregon, Eugene, OR 97405, USA. ✉email: totchy@bu.edu; timg@uoregon.edu

There is growing evidence of the therapeutic benefit of targeted modulation of electrical signaling within the peripheral nervous system (PNS), the network of nerves and ganglia that innervate the organs and tissues of the body[1,2]. Positive clinical trials are reported for vagal nerve stimulation to treat inflammatory diseases[3], depression[4], and epilepsy[5], and beyond these early studies, it is thought that a broad range of chronic diseases may be treatable through precise, artificial control of the PNS[2]. Despite the potential of these bioelectronic therapies, progress—both in our understanding of PNS function and in developing effective modulatory prescriptions—will require new tools capable of making a high-quality, chronically stable interface with nerves. A recently proposed roadmap for the development of such therapies highlights the need for new electrode-based devices that can map peripheral neurophysiology and precisely control end organs[1]. Key features of these emerging peripheral nerve interface (PNI) technologies include stable, high signal-to-noise ratio (SNR) recordings, flexible nerve stimulation options tailorable to physiological effect, and scalable platforms targeting small nerves in preclinical models.

A variety of PNI approaches have been reported, each with unique advantages and disadvantages. Though there are many axes along which PNIs can be distinguished, the position of electrodes relative to nerve fibers is a principal differentiator that correlates with invasiveness, selectivity, safety, and efficacy[6]. Intraneural PNIs—including Utah electrode arrays[7], longitudinal intrafascicular electrodes[8], transverse intrafascicular multichannel electrodes[9], self-opening intraneural electrodes[10], and carbon fiber electrodes[11]—penetrate electrodes through the epineurium to contact individual fascicles and fibers. This proximity yields exceptional recording and stimulating specificity but with substantial risk of trauma to the nerve, disruption of intrafascicular blood flow, and glial scarring. Moreover, many of these devices are too bulky to implant chronically on small nerves. Regenerative PNIs—including sieve electrodes[12], microchannel electrodes[13], regenerative multielectrode interfaces[14], and tissue-engineered electronic nerve interfaces[15]—aim to address these limits by exploiting the ability of peripheral nerves to reestablish connections after transection. Several studies using regenerative PNIs report high-selectivity recording and stimulation over multi-month timescales[16]. However, the need to transect the nerve during implant and endure a prolonged period of compromised function during regrowth limit the range of experiments, therapeutic applications, and nerve targets for which regenerative PNIs are a suitable technology.

Extra-neural PNIs—including split or spiral nerve cuffs[17–19], flat interface nerve electrodes[20], lyse-and-attract cuff electrodes[21], split ring electrodes[22], ribbon electrodes[23,24], and multielectrode softening cuffs[25]—are the most widely used devices targeting the PNS. Their general form is an insulating polymer wrapped around the nerve and containing exposed electrode pads on the interior surface that contact the epineurium; in detail, there is considerable variation among these devices in size, geometry, material composition, mechanical behavior, and fabrication. In contrast to intraneural and regenerative devices, extra-neural PNIs can be implanted without penetrating or transecting the nerve, preserving nerve health but at the cost of reduced electrical access to nerve fibers. Consequently, many extra-neural PNIs have limited recording and stimulating capabilities. (See Table 1 for comparison of extra-neural PNI performance specifications.) More generally, in vivo stability and performance of these devices is limited by several engineering challenges—including poor geometric and biomechanical compliance with the nerve, excessive tethering and compression forces, and scalable fabrication. As such, creating extra-neural PNIs capable of selective

stimulation and high-quality recording over chronic timescales remains a challenge[1,2].

We previously introduced the nanoclip to harness microscale 3D printing to create PNIs with geometries tailored with micron-resolution to the implant target, and we showed that such a device could interface acutely with a small peripheral nerve[26]. Here, we present a nanoclip integrating a multichannel thin-film electrode suitable for chronic nerve interfacing in small animal models, and we demonstrate longitudinally stable, high-SNR recording of behaviorally linked nerve activity. Furthermore, we demonstrate flexible, precision control of an end organ to produce functionally distinct articulatory states supporting stereotyped, fictive vocalizations. Such stable mapping and precision modulation will enable longitudinal studies of PNS function over developmental, disease, and restorative processes and put within reach a new generation of therapeutics centered on closed-loop control of peripheral circuit function[2,27].

## Results

**Peripheral nerve interface design and fabrication.** A recurring challenge restricting chronic viability of PNIs is the mismatch between the biomechanics and scale of implants and those of host tissues. Conventional devices are stiff and bulky, leading to poor contact with the nerve and ultimately degrading performance due to poor compliance with and long-term strain on the tissue[28]. Our approach addresses these concerns by reducing the device scale and limiting engineering tolerances to a few microns. To achieve this, we combine thin-film microfabrication and nanoscale 3D printing[29] to realize an implantable device that matches the scale of small nerves (Fig. 1a). The thin-film electrode array is comprised of a 50 nm layer of gold, defining the electrode pads and interconnects, encapsulated between insulating and biocompatible polyimide layers (Supplementary Fig. 1a). The thickness (12 μm) and narrow width (250 μm) of the device yield a low bending stiffness (~0.5 nN-m), comparable to that of peripheral tissues[30].

The electrode array consists of six gold pads (each $45 \times 80$ μm arranged in a $2 \times 3$ grid; Fig. 1b and Supplementary Fig. 2a) that contact the epineurium of the nerve and are addressed individually by gold interconnects that terminate at input–output (I–O) pads at the opposite end. The I–O pads were connectorized and encapsulated in polydimethylsiloxane for mechanical support and insulation (Supplementary Fig. 1b). Though all six electrodes were within a $260 \times 105$ μm region on the probe head (Fig. 1b), the total length of the thin-film device (from electrode array to I–O pads) was 40 mm—sufficient to reach many peripheral targets in preclinical models while externalizing the connector.

To secure the thin-film array on the nerve, we developed a microscale mechanical anchor—the nanoclip[26]—that consists of two hinged trap doors flanking the entrance of a semi-cylindrical cavity that passes through the body of the device and retains the nerve against the electrodes (Fig. 1a and Supplementary Fig. 2b–d). The nanoclip is fabricated using a two-photon direct-write laser lithography system developed for this application that can print the anchor from standard 3D design files in ~30 s[29]. This enables precise integration of an acrylic photopolymer with the thin-film array without manual assembly steps prone to error and variation. In addition, this fabrication approach allows significant miniaturization (Fig. 1c, d) and the ability to tailor the nanoclip to nerve dimensions with micron-precision, reducing host tissue displacement and ensuring a fit limited only by presurgical measurement of nerve geometry. Following nanoclip printing, iridium oxide films (EIROF) were electrodeposited on the electrodes, reducing impedance and increasing charge storage capacity (Supplementary Fig. 3).

**Table 1 Comparison of extra-neural peripheral nerve interfaces.**

| Investigators | Interface type | Implant site | D (mm) | L (mm) | Experiment | Results |
|---|---|---|---|---|---|---|
| Leob and Peck[61] | Split cuff | Sciatic nerve, cat | 3-4 | 10-15 | Chronic recording, spontaneous | Vpp = 25 µV, 24 days longevity |
| Sahin et al.[62,63] | Spiral cuff | Hypoglossal nerve, dog | 2.5 | 20 | Chronic recording, spontaneous | Vpp = 12 µV, SNR = 2.6 dB 7 months longevity |
| Stocker and Muntzel[17] | Split cuff | Splanchnic nerve, rat | 0.5-1 | 2 | Chronic recording, spontaneous | Vpp = 30 µV, 22 days longevity |
| Sabetian et al.[64] | Split cuff | Sciatic nerve, rat | 1.25 | 13 | Acute recording, stimulation evoked | Vpp = 70 µV, SNR = 5.5 dB |
| Struijk and Thomsen[18] | Spiral cuff | Tibial nerve, rabbit | 2 | 20 | Subchronic recording, natural evoked | Vpp = 10 µV, 5 days longevity |
| Chu et al.[65] | Spiral cuff | Sciatic nerve, rat | 1 | 10 | Acute recording, natural evoked | Vpp = 10 µV, SNR = 3 dB |
| Plachta et al.[66] | Spiral cuff | Vagus, rat | 0.8 | 20 | Acute recording, natural evoked Acute stimulation, single channel | Vpp = 10 µV Imax = 1000 µA, baroreflex activation |
| Lee et al.[67] | Spiral cuff | Sciatic nerve, rat | 1 | 10 | Acute recording, stimulation evoked | Vpp = 3000 µV |
| Tarler and Mortimer[19] | Spiral cuff | Sciatic nerve, cat | 2.7-3.3 | 8 | Acute stimulation, multichannel | Imax = 2000 µA, selective joint torque |
| Tyler and Durand[20] | Flat interface nerve electrode (FINE) | Sciatic nerve, cat | 3-4 | 7 | Acute stimulation, multichannel | Imax = 2000 µA, selective joint torque |
| Elyahoodayan et al.[21] | Lyse-and-attract cuff electrode (LACE) | Sciatic nerve, rat | 1.3 | 10 | Acute recording, stimulation evoked Acute stimulation, single channel | Vpp = 400 µV Imax = 200 µA, EMG recruitment |
| Seo et al.[68] | Neural dust | Sciatic nerve, rat | 1 | 3 | Acute recording, stimulation evoked | Vpp = 900 µV |
| Xiang et al.[24] | Neural ribbon | Sciatic and peroneal nerves, rat | 0.3-0.6 | 3-5 | Acute recording, stimulation evoked | Vpp = 125 µV |
| Lee et al.[22] | Neural clip | Vagus, sciatic, and pelvic nerves, rat | 0.25-0.6 | 2 | Acute stimulation, single channel | Imax = 200 µA, voiding and baroreflex activation |
| González-González et al.[25] | Multielectrode soft cuffs (MSC) | Sciatic and pelvic nerves, rat | 0.2-1 | 1 | Acute recording, natural evoked Subchronic recording, stimulation evoked Acute stimulation, multichannel | Vpp = 150 µV Vpp = 400 µV, 30 days longevity Imax = 100 µA, selective joint angle |
| Present work | Nanoclip, polyimide thin-film electrodes | Tracheosyringeal nerve, zebra finch | 0.15 | 0.3 | Acute recording, stimulation evoked Chronic recording, spontaneous Acute stimulation, multichannel | Vpp = 3000 µV, SNR = 48.1 Vpp = 500 µV, SNR = 10, 30 days longevity Imax = 200 µA, fictive singing |

D estimates the diameter of the implanted nerve. L indicates the length of the nerve interface. Stimulation evoked: recorded activity was evoked by electrical stimulation of the implanted nerve. Natural evoked: recorded activity was evoked by naturalistic activation of sensory responses. Spontaneous: recorded activity was result of spontaneous behavior or processes.

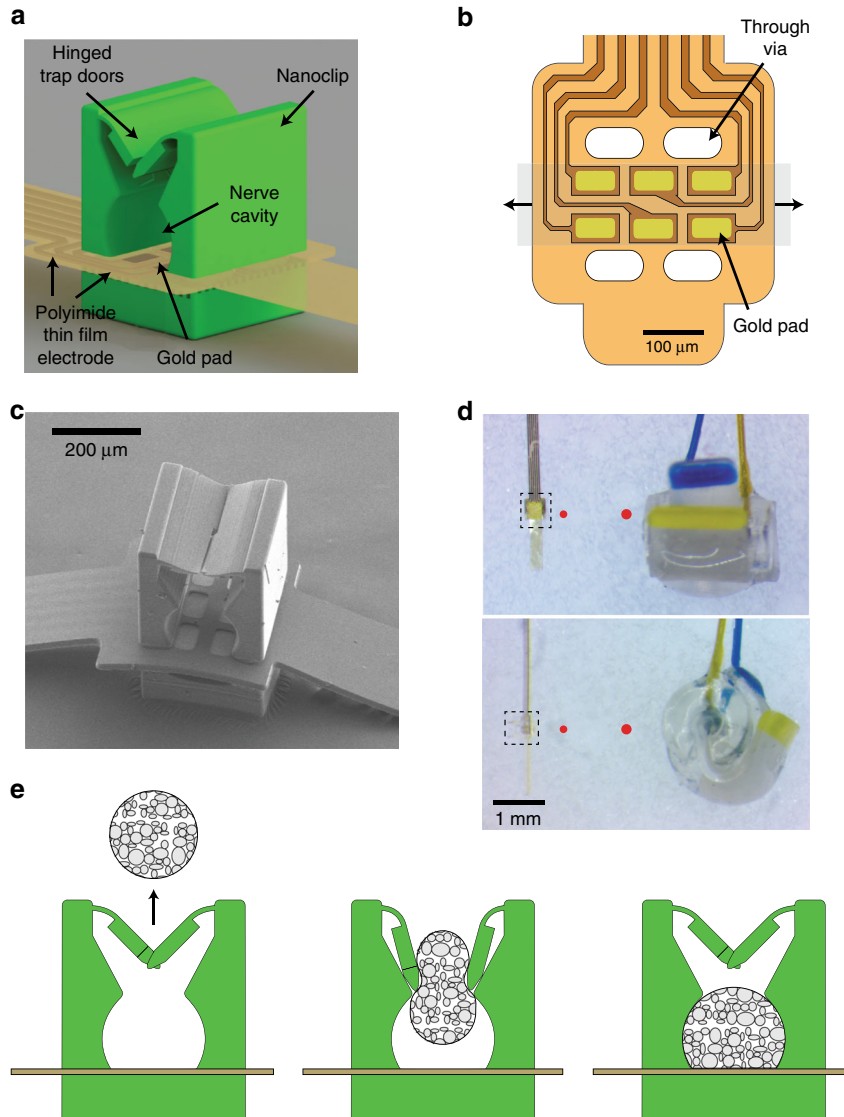

**Fig. 1 Thin-film-integrated nanoclip nerve interface overview. a** Rendering of a thin-film-integrated nanoclip nerve interface showing key components. The nanoclip was fabricated directly on the thin-film electrode via rDLW[29] and consists of two interlocking trap doors with elastically deformable hinges, a nerve retention cavity, and a rigid base. **b** Diagram of the polyimide thin-film electrode array used in all experiments. Six gold electrode pads (45 × 80 μm each) were located inside the nerve cavity (in (**a**)) such that they were in contact with the epineurium of the retained nerve. The thin-film through-vias allow for the mechanical integration of the nanoclip top and base (see Supplementary Fig. 2). The shaded gray region shows the location of a retained nerve; the arrows indicate the nerve axis. **c** SEM micrograph of the nanoclip nerve interface. **d** Micrograph of the nanoclip (left) and a Cortec silicone nerve cuff (right) sized for 150 and 200 μm nerves, respectively. Top image taken from overhead view; bottom shows the same devices in profile. Dashed bounding boxes highlight the printed anchor and approximate the region shown in (**c**). For comparison, the red circles show the diameter of the nerve for which each device is designed (i.e., left: 150 μm; right: 200 μm). **e** Schematic of implantation process: (left) the nanoclip is advanced toward an isolated nerve; (center) the nanoclip contacts the nerve resulting in elastic deformation of the trap doors and subsequent entry of the nerve into the central retention cavity; (right) as the nerve fills the cavity and contacts the electrode pads, the doors close behind the nerve to secure the nanoclip.

**Uncompromised nerve function following implant**. Recent studies suggest that some PNIs are associated with compromised nerve function at acute or chronic time points[31,32], casting doubt on prior findings establishing basic PNS physiology and the feasibility of bioelectronic technologies. Thus, a principal design constraint for the nanoclip was to ease surgical manipulation of the device and nerve during implant such that insult to the host tissues would be minimal. For implantation, the nanoclip is advanced toward the nerve (Fig. 1e, left), elastically deforming the trap doors (Fig. 1e, center) and permitting entry of the nerve into the central retention cavity. After the nerve clears the doors, they return to their original configuration, irreversibly securing the nanoclip on the nerve (Fig. 1e, right). To estimate the forces on

the nerve during implant, we performed finite element mechanical modeling of the trap doors using upper- and lower-bound estimates for the photoresist mechanical properties (Supplementary Fig. 4a, b). These simulations suggest that implant forces (i.e., forces required to fully open the trap doors) are in the range of 1.25–7.5 μN (Supplementary Fig. 4c–e). Several studies indicate that 30 mmHg is an important upper limit for compression before onset of nerve damage and compromise of signaling[33]. Given the size of the nanoclip and the small surface area in contact with the nerve during implant (~0.03 mm²), this corresponds to an upper force limit of ~120 μN—1.5–2 orders of magnitude greater than our simulations estimate are required. Furthermore, this analysis estimated that under full loading

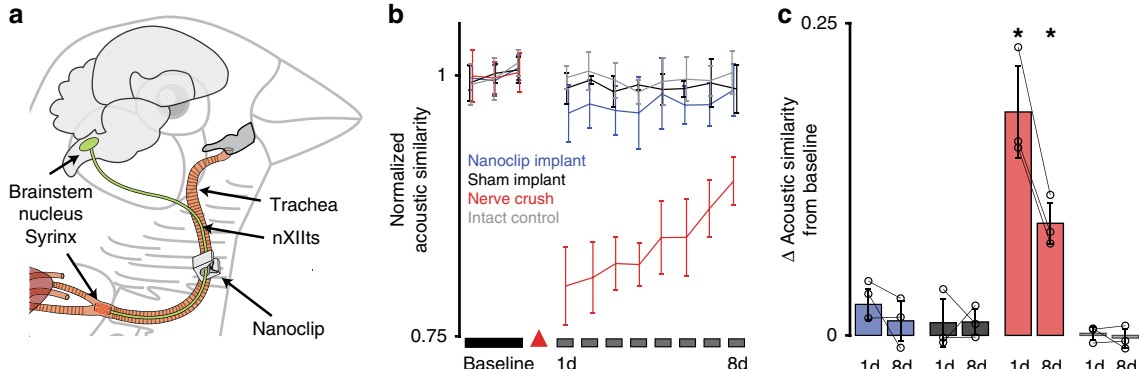

**Fig. 2 Nanoclip nerve interface chronic implantation and functional safety. a** Diagram of the nanoclip interface implanted on the nXIIts nerve of the zebra finch. **b** Functional assessment of nXIIts following bilateral nanoclip implant (blue), sham implant (black), nerve crush (red), or intact (gray) quantified as the acoustic similarity of baseline to post-manipulation song structure (see "Methods"). Line and error bars denote mean ± Std across n = 3 birds per condition. Red triangle marks day of surgery. **c** Summary statistics showing the difference in acoustic similarity between baseline song and songs produced on days 1 and 8 post manipulation. Colors denote experimental conditions as in (**b**). Each data point represents the mean across all motifs from a single bird. Bars and error bars denote mean ± Std across n = 3 birds per condition. Two-tailed paired *t*-test, day 1: P = 0.02; day 8: P = 0.016. *P < 0.05. Source data are provided as a Source Data File.

conditions maximal stresses within the door hinges were well below the elastic and fatigue limits of the materials (Supplementary Fig. 4f, g). Taken together, this analysis is consistent with the small device being sufficiently robust to withstand the biomechanical forces of implantation and body dynamics.

The nanoclip is sized for a snug but not tight fit, with complete closure of the trap doors and minimal chronic compression of the nerve (see Supplementary Fig. 2b–d for device dimensions). During implant surgery and acute experiments, no spontaneous movement of the device relative to the nerve was observed. Though a microsuture may be used to stabilize the interconnect within the body to minimize potential host tissue damage, no additional anchoring methods or surgical adhesives were required to secure the device irreversibly on a small nerve. In all chronically implanted animals for which necropsies were performed (n = 5 birds), the nanoclips were found securely latched over the nerve up to 11 months after implant, suggesting the potential for long-term viability.

To evaluate the chronic functional safety of the design, we implanted the nanoclip on the songbird tracheosyringeal nerve (nXIIts)—an avian hypoglossal analog that innervates the songbird vocal organ, the syrinx (Fig. 2a), and shows strong homologies to mammalian sensorimotor nerves, containing ~1000 myelinated and unmyelinated fibers[26]. We reasoned that any disruption of nerve function due to the surgery or implant would be revealed in acoustic distortion of otherwise highly stereotyped song[26]. We recorded song from adult male zebra finches (n = 3 birds) for 3 days before and 8 days after bilaterally placing nanoclips on the nXIIts. In all experiments, birds resumed singing the day after surgery, robustly producing well-structured vocalizations from the first utterances. (See Supplementary Fig. 5 for representative examples of baseline and post-implant songs.) To quantify changes in nerve function following implant, we calculated the moment-by-moment acoustic similarity of pre-implantation songs to songs produced in subsequent days (Fig. 2b). We found a small but not significant difference in song acoustic structure produced before and after 1 and 8 days post implant (P = 0.073 and 0.417, respectively; n = 3 birds; see "Methods" for details of all statistical analyses)—consistent with changes measured in birds receiving bilateral sham implants (P = 0.53 and 0.28; n = 3 birds) and intact controls (P = 0.70 and 0.73; n = 3 birds) (Fig. 2c). For comparison, birds receiving bilateral nerve crush injuries showed large changes from baseline in song

acoustic structure (P = 0.02 and 0.016; n = 3 birds), that were significantly different from those of nanoclip, sham, and intact experimental groups (P > 0.22). In total, these simulations and in vivo experiments establish the functional safety of the nanoclip for chronic implant on small nerves.

**High-SNR acute recordings from fine peripheral nerves**. To validate the nanoclip for in vivo recording of nerve activity, we recorded evoked compound responses from the nXIIts in anesthetized zebra finches. The nXIIts was exposed by blunt dissecting the tissue surrounding the trachea, and the PNI was implanted on an isolated ~1 mm section of the nerve. For bulk stimulation, silver bipolar hook electrodes were placed on a similarly isolated section of the nXIIts ~15–20 mm distally (Fig. 3a). Biphasic stimulation pulses (200 µs phase$^{-1}$) were applied at 1 Hz, and voltages were recorded from 5 ms before to 25 ms after the stimulation onset.

We obtained graded evoked response curves by varying the stimulation current amplitude (Fig. 3b). Consistent with estimates of nerve conduction velocities in myelinated axons of a similar size[34], the most salient features of the evoked responses occurred ~0.75 ms poststimulation and persisted for up to 4 ms. Across experiments (n = 3 birds), the peak-to-peak voltages (Vpp) of evoked responses showed a sigmoidal relationship with stimulation intensity (Fig. 3c), as expected[35]. The minimal evoked response detected by the nanoclip was 37.6 ± 6.5 µV at 14.2 ± 5.4 µA with an SNR of 8.4 ± 0.95 dB; at the mean stimulation intensity assayed (i.e., 60 µA), evoked responses were recorded with an SNR of 48.1 ± 4.2 dB. To confirm that recorded responses were of neuronal origin, we blocked nerve transduction with lidocaine (2.0% in saline) applied at the stimulating site (Fig. 3d). Across experiments (n = 3 birds), evoked response amplitudes following nerve block were significantly different from saline controls (P = 6 × 10$^{-6}$); subsequent washout restored response amplitudes to not significantly different from control (P = 0.15; Fig. 3e).

**Stable mapping of behaviorally linked small nerve activity**. A principal goal in developing the nanoclip was to realize a device capable of making high-quality recordings of small nerve activity over the multi-week durations relevant for studying developmental, disease, and restorative processes. In adult zebra finches,

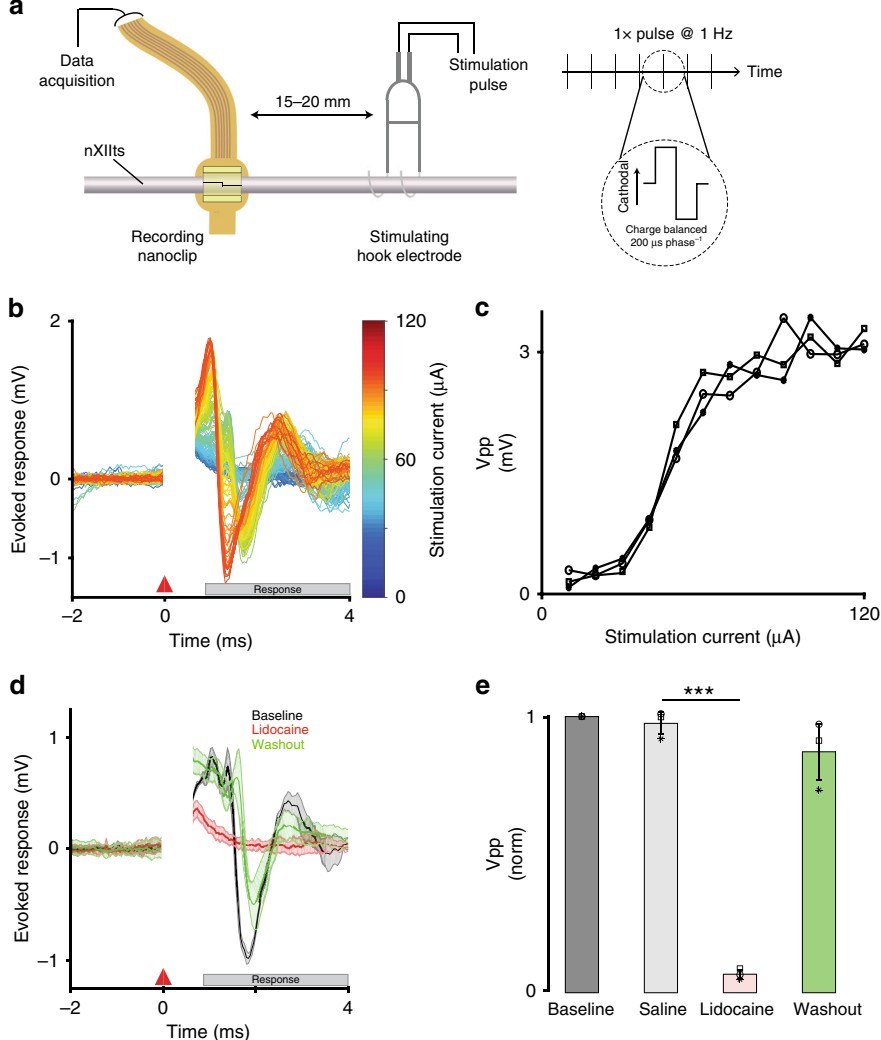

**Fig. 3 Nanoclip-anchored thin-film arrays produce high-quality electroneurograms in acute preparations. a** Schema for acute recording of evoked compound action potentials. (Left) Current-controlled stimulation was delivered via bipolar silver hook electrodes; evoked responses were recorded by a nanoclip interface implanted 15–20 mm rostrally. (Right) Biphasic, cathodal-leading stimulating pulses, 200 μs phase$^{-1}$ at 10–120 μA, were delivered at 1 Hz for 16–24 trials. **b** Example of graded evoked response to increasing stimulation intensities. Stimulation applied at $t = 0$ ms; stimulation artifact appears $t = 0$–0.5 ms; evoked response follows. **c** Evoked response peak-to-peak voltage (Vpp) showed an expected sigmoidal relationship with stimulation intensity. Each data point indicates the mean across trials within an animal ($n = 3$ birds; 16–24 trials each per stimulation intensity). Symbols identify individual birds. **d** Example of stimulation evoked responses recorded before, during, and after local lidocaine application. Each condition shows mean ± SEM (shaded) for $n = 16$ trials; all responses evoked at 64 μA. **e** Evoked response peak-to-peak voltage for different experimental conditions. Response amplitudes were normalized to baseline condition to facilitate comparison across $n = 3$ animals. Each data point indicates the mean across 16 trials within an animal; symbols identify individual birds. Bars and error bars denote mean ± SEM across $n = 3$ birds per condition. Repeated-measures ANOVA, $P = 0.003$; Dunnett's test $P = 6 \times 10^{-6}$. ***$P < 0.001$. Source data are provided as a Source Data File.

the motor program underlying crystallized song is composed of highly stereotyped patterns of cortically generated neuronal activity that are precisely time locked to song and stable over multi-month periods of time[36]. Thus, to assess the long-term stability and reliability of the interface, we implanted the nanoclip on the nXIIts, the primary output of the central nervous system (CNS) song system and the sole source of innervation to the syrinx, and recorded singing-related nerve activity from freely behaving birds ($n = 5$; Fig. 4a).

As in recordings from singing-related CNS structures[36,37], we observed robust multi-unit activity with large amplitude modulations during singing (Fig. 4b, top) but less so during movements so vigorous as to produce detectable cage sound (Fig. 4b, red band). The large multi-unit signal survived common-mode subtraction and filtering (Fig. 4b, middle). Consistent with the

prior studies of singing-related activity in the CNS, we found that stereotyped segments of song (Fig. 4b, gray bands) were reliability associated with signal envelopes exhibiting similar time-varying trajectories (Fig. 4b, bottom).

In $n = 3$ animals, we recorded singing-related nXIIts activity for more than 30 days (31, 34, and 37 days; in two other animals recording durations were 17 and 19 days, see "Methods"). Representative recordings at 1, 10, 20, and 30 days post implant showed similarly well-defined signals with modulation amplitudes of up to 500 μV (Vpp across days 1, 10, 20, and 30: 438 ± 64 μV; range: 334–522 μV; $n = 3$ birds; Fig. 4c). These recordings showed a remarkable degree of stereotypy in song-aligned activity envelopes over the 30-day period (Fig. 4c, d). To quantify the stability of these recordings over time, we calculated trial-by-trial (1) the Pearson correlation between song-aligned nXIIts activity

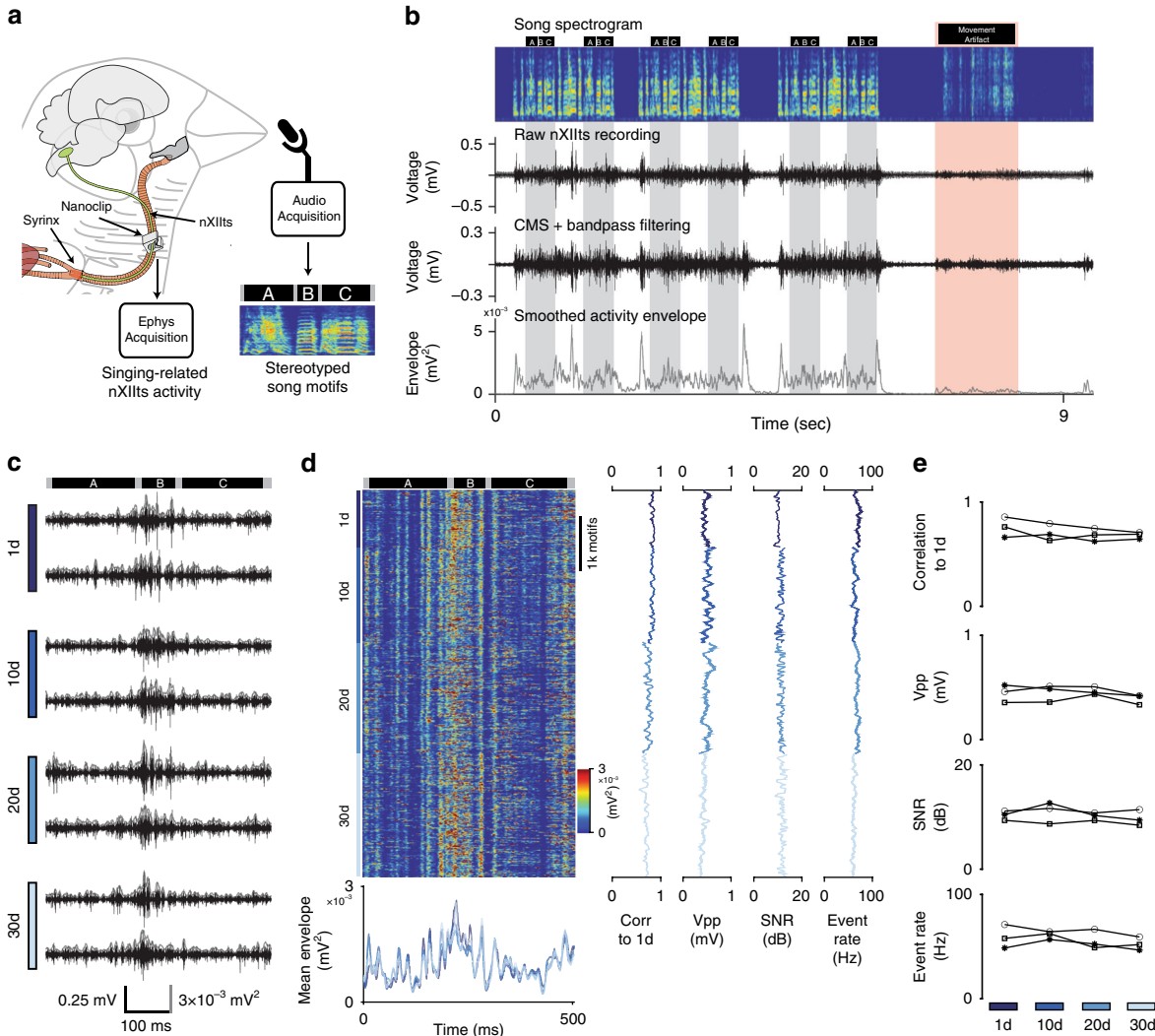

**Fig. 4 Long-term stable recordings with minimal degradation over more than 4 weeks. a** Schema for chronic recording of nXIIts activity in the singing bird. A nanoclip was implanted on the right-side nXIIts; song-triggered acquisition software captured both vocalization and concurrent nXIIts activity. Example song motif spectrogram from bird shown in (**b–d**); black bars above the spectrogram indicate the beginning and end of each syllable in the motif ("A," "B," and "C"). **b** Representative example of chronic nXIIts recording aligned to song. Top row: spectrogram of bird's song containing three song bouts, each with two repeated motifs. Black bars above the spectrogram indicate song syllable boundaries; gray bands denote song motifs. Red band identifies movement-related noise and associated motion artifact in the electrophysiology recording. Second row: unprocessed nXIIts activity. Third row: nXIIts activity (from second row) after common-mode subtraction (CMS) and filtering. Bottom row: smoothed nXIIts activity envelope. **c** Representative nXIIts recordings from the bird are shown in (**b**) on days 1, 10, 20, and 30 post implant. Gray line shows the smoothed activity envelope for each recording. Dark bars at top indicate the recording alignment to syllables; bars at bottom indicate time, voltage, and signal envelope scale. **d** Stability of chronic recordings from the bird appearing in (**b, c**) over 30 days. Left-top: song-aligned nXIIts activity envelopes for motifs produced on days 1, 10, 20, and 30 post implant. Left-bottom: mean song-aligned activity envelope (over ten trials) recorded on days 1, 10, 20, and 30 post implant. Line colors indicate recording day. Middle left: running correlation between the song-aligned nXIIts activity envelope and the mean day 1 activity envelope. Middle: running peak-to-peak voltage during singing. Middle right: running SNR of recordings during singing. Far-right: Running estimated event rates during singing. **e** Summary data of chronic recordings over 30 days in $n = 3$ birds; symbols identify individual birds. Top: mean daily trial-by-trial Pearson's correlation to the average activity pattern on the 1st day of singing. Middle-top: mean daily peak-to-peak voltage. Middle-bottom: mean daily SNR of nXIIts recordings. Bottom: mean daily event rates. Source data are provided as a Source Data File.

envelopes and the mean activity envelope on day 1, (2) the Vpp of singing-related activity, (3) the SNR of singing-related activity, and (4) the event rate of singing-related activity (Fig. 4d, right). Across $n = 3$ birds, we found no significant differences in the daily means of these stability metrics between day 1 and any subsequent day (correlation: $P > 0.089$; Vpp: $P > 0.156$; SNR: $P > 0.208$; event rate: $P > 0.147$; Fig. 4e). These analyses demonstrating stable longitudinal recordings of nXIIts activity suggest that the nanoclip is well-suited to mapping PNS function at chronic timescales.

**The neuronal origin of chronically recorded signals**. To a greater degree than CNS interfaces, chronic PNIs must endure vigorous body dynamics that can not only displace implanted devices but also generate movement-related recording artifacts (Fig. 4b, red band). In addition, the proximity of implanted nerves to muscles can make EMG invasion of PNS recordings a significant concern. In our chronic nXIIts recording preparation (Fig. 4a), the nanoclip implant site is ~5–10 mm from the nearest skeletal muscles (the multifidus cervicis in the neck), >25 mm from the syrinx, and 1–2 mm from the smooth muscle at the

posterior trachea. Virtual referencing and processing techniques, like common-mode subtraction (Fig. 4b, middle), are widely used to attenuate correlated noise sources on multichannel recordings[38]. Nevertheless, given several plausible sources of invasion and the absence of prior reports against which to compare our nXIIts recordings (Fig. 4), we sought to establish directly the neuronal origin of nanoclip-recorded signals.

To assess the degree to which our nXIIts recordings reflect neuronal versus nonneuronal sources, we made simultaneous recordings from nanoclip interfaces with electrodes in contact with (on nerve) and adjacent to (off nerve) the nXIIts. For off-nerve recordings, we printed a second nerve anchor on the underside of a nanoclip such that the interface could be implanted on the nXIIts while holding the electrode ~150 µm from the nerve surface with the pads facing away from the nerve and exposed to the extraneural environment (Fig. 5a). The on-nerve and off-nerve interfaces were implanted on the same nerve ~5 mm apart; given the anatomy, the two nanoclips were approximately equidistant from the proximal muscles. Thus, we reasoned that each PNI would be subject to the same biomechanical and electrochemical (i.e., nonneuronal) artifact sources, but only the on-nerve recordings would reflect the additional contribution of nXIIts neuronal activity at the surface of the electrode pads. In $n = 2$ birds, we recorded with on-nerve and off-nerve devices voltage signals that were temporally correlated with singing (Fig. 5b, top). As in the single-nanoclip recording experiments (Fig. 4), on-nerve recordings showed well-defined signals with large amplitude fluctuations; however, off-nerve recordings were poorly defined and of consistently smaller amplitude. Strikingly, only in on-nerve recordings did these singing-related fluctuations survive common-mode subtraction (Fig. 5b, bottom; see also Supplementary Fig. 6a, b). This suggests that though there may be nonneuronal sources contributing to the nanoclip-recorded signals in this preparation, their effect is correlated across the six channels making it possible to subtract cleanly the nonneuronal signal by virtual referencing. To quantify the relative strength of the neuronal and nonneuronal components of the recordings, we calculated the trial-by-trial SNR of singing-related signals from each device (Fig. 5c and Supplementary Fig. 6c), using singing-free epochs within each recording to estimate the in vivo noise floor (Fig. 5b, yellow band). Across animals, we found large differences in the SNR of on-nerve and off-nerve signals ($P = 0.25$, $n = 2$; Fig. 5e, Supplementary Fig. 6d). To quantify the temporal relationship between the recordings, we calculated the trial-by-trial correlation between the singing-related on-nerve and off-nerve signals (Fig. 5d). Across animals, we found these correlations were not significantly different from 0 ($P = 0.13$, $n = 2$; Fig. 5e). Thus, on the small scale of the nanoclip, the six recording contacts show highly correlated movement and EMG artifacts and therefore can be cleanly subtracted. In contrast, over the same length scale most neural signals are much less correlated across electrodes and survive the common-mode subtraction. Taken together, these experiments suggest that the extent of artifact invasion in our recording preparation is limited and are consistent with on-nerve nanoclip-recorded signals principally reflecting neuronal activity.

**Precision control of a small nerve to evoke fictive behavior.** Precise in vivo stimulation of peripheral nerves is essential for probing function and for providing therapeutic control of limbs and end organs[1,27]. Using a two-nanoclip experimental preparation, we recorded compound responses with a rostrally placed nanoclip that were evoked with a second interface implanted 15 mm caudally (Fig. 6a). Biphasic stimulation pulses (200 µs phase$^{-1}$) were applied at 1 Hz, and data were recorded for 5 ms before and 25 ms after stimulation onset. We obtained

graded evoked responses by varying the stimulation current (Fig. 6b). Across experiments ($n = 3$ birds), we found stimulation thresholds of $9.2 \pm 1.6$ µA, consistently lower than those from bipolar silver hook stimulation electrodes (at $14.2 \pm 5.4$ µA; Fig. 3). These lower stimulation thresholds are consistent with the nanoclip providing substantial isolation of the electrodes from the surrounding extra-neural environment, thus reducing current leakage through the device and increasing the efficiency of tissue depolarization.

Having demonstrated the capacity for bulk stimulation, we wondered whether precisely steering stimulation currents between the six electrodes would enable access to a wider range of nerve activation states. Using the two-nanoclip preparation (Fig. 6a), we recorded evoked compound responses evoked using spatially distinct multichannel stimulation patterns (biphasic pulses, 200 µs phase$^{-1}$ @ 1 Hz). We found that responses with unique waveforms and amplitudes could be evoked by changing the spatial stimulation pattern while keeping total current flow constant across trials (Fig. 6c). This suggested that, despite the small scale of the nanoclip, multichannel, current-steering-based stimulation can nevertheless achieve modulatory specificity over that of bulk stimulation.

Though differences in the waveforms of the current-steering-evoked responses were visually apparent (Fig. 6c), it remained unclear whether such differences in nXIIts activation were of functional significance. To test whether the nanoclip was capable of precise functional modulation, we developed a surgical preparation —fictive singing—that provides complete control over the vocal-respiratory system of the anesthetized songbird. As in human speech, zebra finch vocalization requires both mechanical control of the vocal organ and elevated air pressure. Hence in the fictive singing paradigm, the muscles of the syrinx are articulated by multichannel stimulation of the nXIIts while the respiratory system is pressurized externally. Physiological levels of air pressure during natural singing[39] were maintained via a cannula placed in the abdominal air sac (Fig. 7a). In this preparation, we found that unilateral stimulation of the nXIIts (100 biphasic pulses, 200 µs phase$^{-1}$ @ 1 kHz) was sufficient to reliably transition the vocal system from quiescence, to a sound-producing state, and back to quiescence (Fig. 7b). This demonstrates the nanoclip capable of stimulating small nerves to a physiologically efficacious state.

To functionally validate the nanoclip for precision nerve control, we quantified differences in spectral structure of fictive vocalizations evoked by spatially distinct patterns of multichannel stimulation. We found that repeated application of a single stimulation pattern could reliably produce stereotyped fictive vocalizations with spectrotemporal characteristics consistent with species-typical song elements. Furthermore, we observed that spatially inverted current-steering patterns often elicited fictive vocalizations that were acoustically distinct from those produced by the original pattern (Fig. 7b). To visualize differences in the acoustic structure of fictive vocalizations produced by a set of 24 distinct stimulation patterns (Supplementary Fig. 7), we embedded high-dimensional representations of fictive vocalizations (i.e., high-resolution spectrograms) in two-dimensional space using t-distributed stochastic neighbor embedding (t-SNE). We found that this dimensionality reduction produced distinct clusters that corresponded to variations in stimulation patterns (Fig. 7c). To quantify the reliability and specificity of nanoclip multichannel stimulation, we compared the mean pairwise acoustic similarity between each vocalization produced by a given stimulation pattern against the mean pairwise acoustic similarity between vocalizations produced by different stimulation patterns. Across experiments ($n = 6$ birds, 24 patterns of ~20 trials each), we found that acoustic similarity within stimulation patterns was significantly greater than across pattern similarities

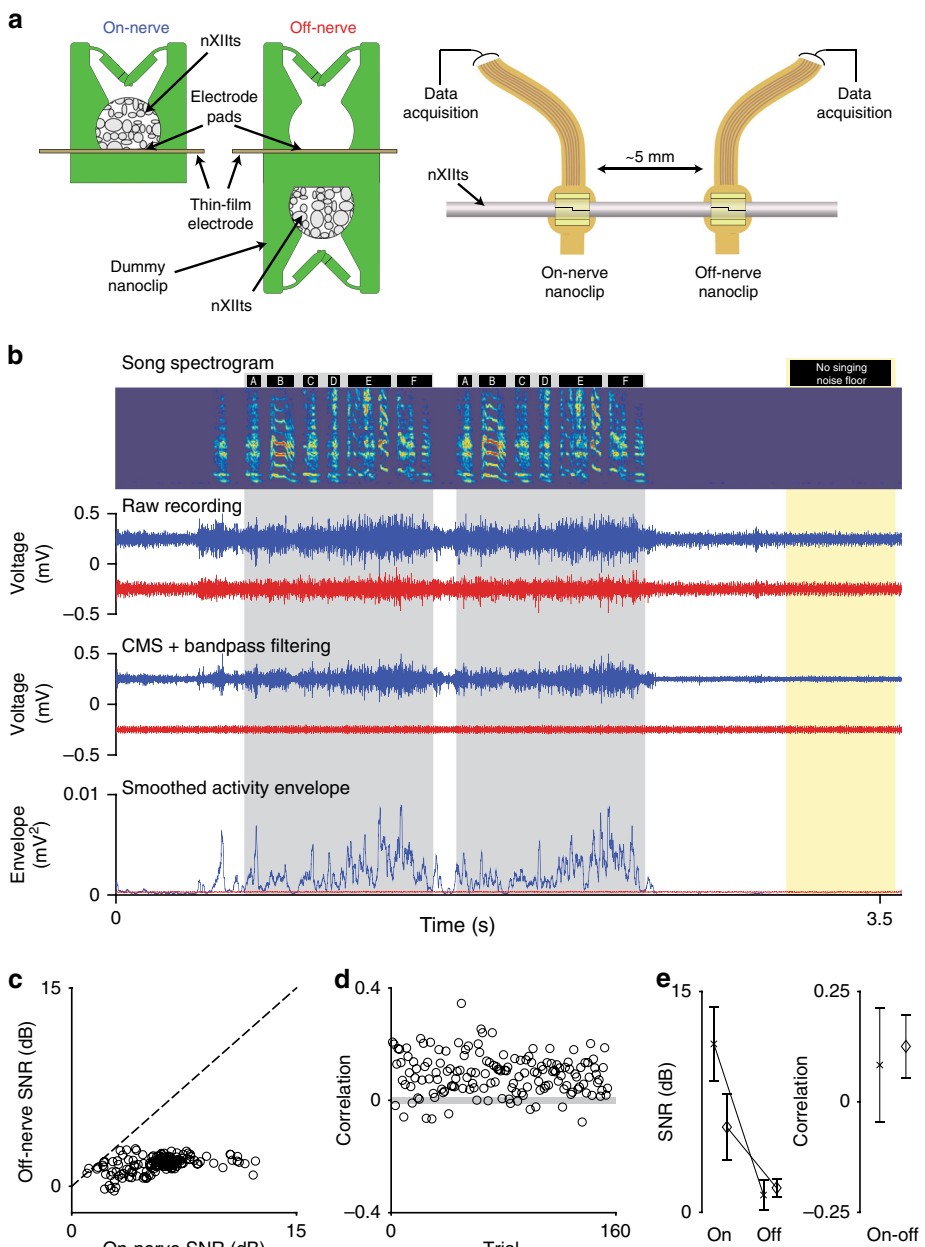

**Fig. 5 Experiments addressing the neuronal origin of recorded signals. a** Schema for simultaneous two-nanoclip chronic recordings. (Left) On-nerve recordings were made with the standard nanoclip design (used in all other experiments) in which the implanted nerve rests within the cavity and in contact with the electrode pads. Off-nerve recordings were made using a modified device with a second "dummy" nanoclip on the underside of the interface, keeping the electrode pads out of contact with the nerve and able to capture nonneuronal signals. (Right) nXIIt activity was recorded via an on-nerve nanoclip; control recordings were made via an off-nerve nanoclip interface placed ~5 mm caudally. **b** Representative example of simultaneous on-nerve and off-nerve chronic recordings. Top row: spectrogram of bird's song containing two repeated song motifs. Black bars above the spectrogram indicate syllable boundaries; gray bands denote song motifs. Yellow band identifies song and artifact-free section of the recording from which the noise floor was estimated for SNR. Second row: unprocessed signals from on-nerve (blue) and off-nerve (red) interfaces. Third row: on-nerve (blue) and off-nerve (red) signals (from second row) after CMS and filtering. Bottom row: smoothed envelope for on-nerve (blue) and off-nerve (red) signals. **c** Trial-by-trial comparison of SNR for on-nerve and off-nerve signals for the bird is shown in (**b**). Each circle represents an individual trial; dashed line indicates unity. **d** Trial-by-trial Pearson correlations between the on-nerve and off-nerve envelops for the bird are shown in (**b**). Each circle represents an individual trial; the gray-shaded box indicates the 95% CI for zero-correlation by bootstrap. **e** Summary of neuronal and nonneuronal chronic recordings from the nXIIts in two birds. (Left) SNR of on- and off-nerve signals. (Right) Correlation of singing-related signal envelopes for on- and off-nerve signals. For each plot, symbols indicate the mean across motifs for each animal; error bars indicate Std ($n = 154$ and $56$ motifs, respectively). Source data are provided as a Source Data File.

($P = 3.5 \times 10^{-5}$; Fig. 7d). For comparison, we performed the same within- and across-type acoustic similarity analysis on naturally produced vocalizations, finding that the fictive experiment well replicated the reliability and specificity with which the vocal apparatus is controlled during natural singing ($P = 0.0016$, $n = 6$

birds; Fig. 7d). These experiments suggest that each current-steering pattern activated distinct subsets of nerve fibers and could do so reliably across multiple renditions. Furthermore, it suggests that in this preparation the nanoclip is capable of producing nerve activity states that are on average at least as

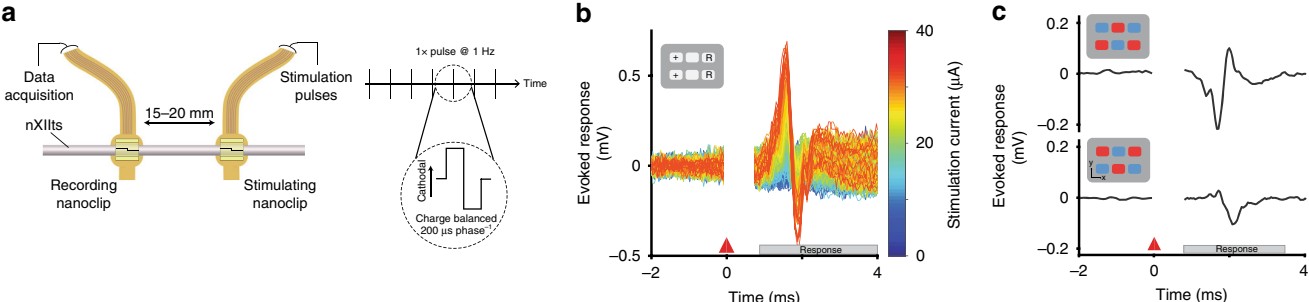

**Fig. 6 Nanoclip-anchored thin-film arrays enable low-threshold bulk and multichannel nerve stimulation. a** Schema for acute stimulation of evoked compound action potentials. (Left) Current-controlled stimulation was delivered via a rostral nanoclip interface; evoked responses were recorded by a second nanoclip interface placed 15–20 mm caudally on the same nerve. (Right) Biphasic, cathodal-leading stimulating pulses, 200 μs phase$^{-1}$ at 5–40 μA, were delivered at 1 Hz. **b** Example of graded evoked response to increasing stimulation intensities. Stimulation applied at $t = 0$ ms; stimulation artifact appears $t = 0$–0.5 ms; evoked response follows. Inset shows stimulation electrode configuration—"+" denotes source of cathodal first phase; "R" denotes electrodes for the current return path. **c** Evoked compound responses show that spatially distinct patterns of stimulation can produce a high degree of modulatory specificity. Inset shows the location of electrodes mediating stimulation pulses with cathodal (red) or anodal (blue) leading phases.

dissimilar from each other as those occurring in the context of natural singing. In total, this validates the nanoclip for precision functional modulation of small nerve activity.

## Discussion

The development of new devices for detection and modulation of signaling patterns within the nervous system puts within reach a new class of bioelectronic therapies[1,2]. The PNS will be at the center of these treatments, as the range of functions it controls in chronic disease suggests many therapeutic targets, improving the lives of large patient populations. Furthermore, the relative surgical accessibility (compared to intracranial structures) and hierarchical organization of the PNS, whereby branches diverge to become more functionally specific and homogenous, provide an opportunity for tractable, targeted modulation of end organs with minimal off-target side effects. For these approaches to become viable, PNIs that not only nominally fit the implant target, but also match the overall scale, geometry, and biomechanics of nerves are necessary. However, conventional PNIs have geometries and mechanical properties orders of magnitude divergent from those of the smallest nerve targets, increasing risk of insult to host tissues and diminishing efficacy due to poor compliance.

To address these challenges, we developed the nanoclip—a microscale nerve interface (Figs. 1 and 2) fabricated using a custom direct laser writing system that prints the anchor from standard CAD files in ~30 s. With this printing speed and digital design, it is feasible to manufacture nanoclips with a range of sizes and shapes to achieve optimal fit. We show that precisely sized nanoclips can achieve stable, high-SNR recordings from a small nerve over multi-week timescales (Figs. 3–5). Over the full time-course of recordings, neural signals at the hundred microvolt scale and above were recorded—a signal quality that is typically only associated with acute preparations (Table 1). In addition, we demonstrated the nanoclip can achieve a precise functional modulation of the songbird syrinx (Figs. 6 and 7). To our knowledge, a chronic format PNI with comparable performance has not been described previously. This device provides a new reference for safe implant, stable high-SNR recording over multi-week timelines, and precise modulation of small peripheral nerves.

Still, the nanoclip performance does not arise from any novel insight or change in interfacing mode; this extra-neural PNI fundamentally resembles devices used for decades. The difference, we believe, is a combination of factors, all of which may be important. The design, manufacturing tolerances, and fit-to-nerve

contribute to a mechanistic hypothesis of improved recording signal quality: by closely fitting the device to the nerve, a high-conductivity path (provided by saline) between the electrodes and the external environment has been reduced. Though not directly tested here, this hypothesis is consistent with prior studies demonstrating that reducing conductivity through a nerve cuff can significantly increase Vpp[40,41]. The importance of the electrode-tissue interface for signal quality is also seen elsewhere—the loose patch or juxtacellular recording methods using glass pipette electrodes. There, microvolt-scale extracellular recordings transition to millivolt scale signals as a glass pipette approaches a cell and forms a high-impedance seal with the cell membrane[42]. In addition, we speculate from prior studies that the small device size, highly flexible interconnect, and snug fit may slow or reduce local reactive tissue responses that over time can grow between the recording electrode and the nerve, degrading interfacing efficacy[43–45].

Future studies in additional species and nerves will be necessary to further validate the potential of the nanoclip for high-SNR chronic recording. It is possible that the nerve recorded here, the nXIIts, demonstrates voltage fluctuations on the surface that are larger than in other common targets such as the sciatic, splanchnic, or pelvic nerves. As the signals recorded with extra-neural electrodes are (in part) a function of the super position of nearby active current sources within the nerve[46], both the number of co-active nerve fibers and their firing frequency can affect the amplitude of recorded signals. Sciatic, pelvic, and splanchnic fiber recruitment vary widely by assay, but single fiber firing rates are reported <50 Hz[47–49]. We are not aware of prior studies characterizing nXIIts activity, but recordings from other singing-related brain regions suggest very high firing rates (>200 Hz) and a large fraction of co-active units during singing are common[50,51]. Thus, it is possible that high firing rates or synchronous activity in the nXIIts contribute to unusually large multi-unit potentials on the nerve surface. Nevertheless, the key observation is not that the nanoclip yields unusually large signals—indeed, they are comparable to those reported in some acute preparations (Table 1)—but rather that these signals remain stable over the timescale of weeks. Speculation about high firing rates or synchrony does not provide an explanation for stable performance. Instead, we believe the best explanation is that few chronic recordings from small nerves have been reported previously, and that with devices precisely tailored to their size and geometry, high-SNR signals may be more accessible than has been anticipated. If true, this is good news for the study of small nerve signaling in the periphery.

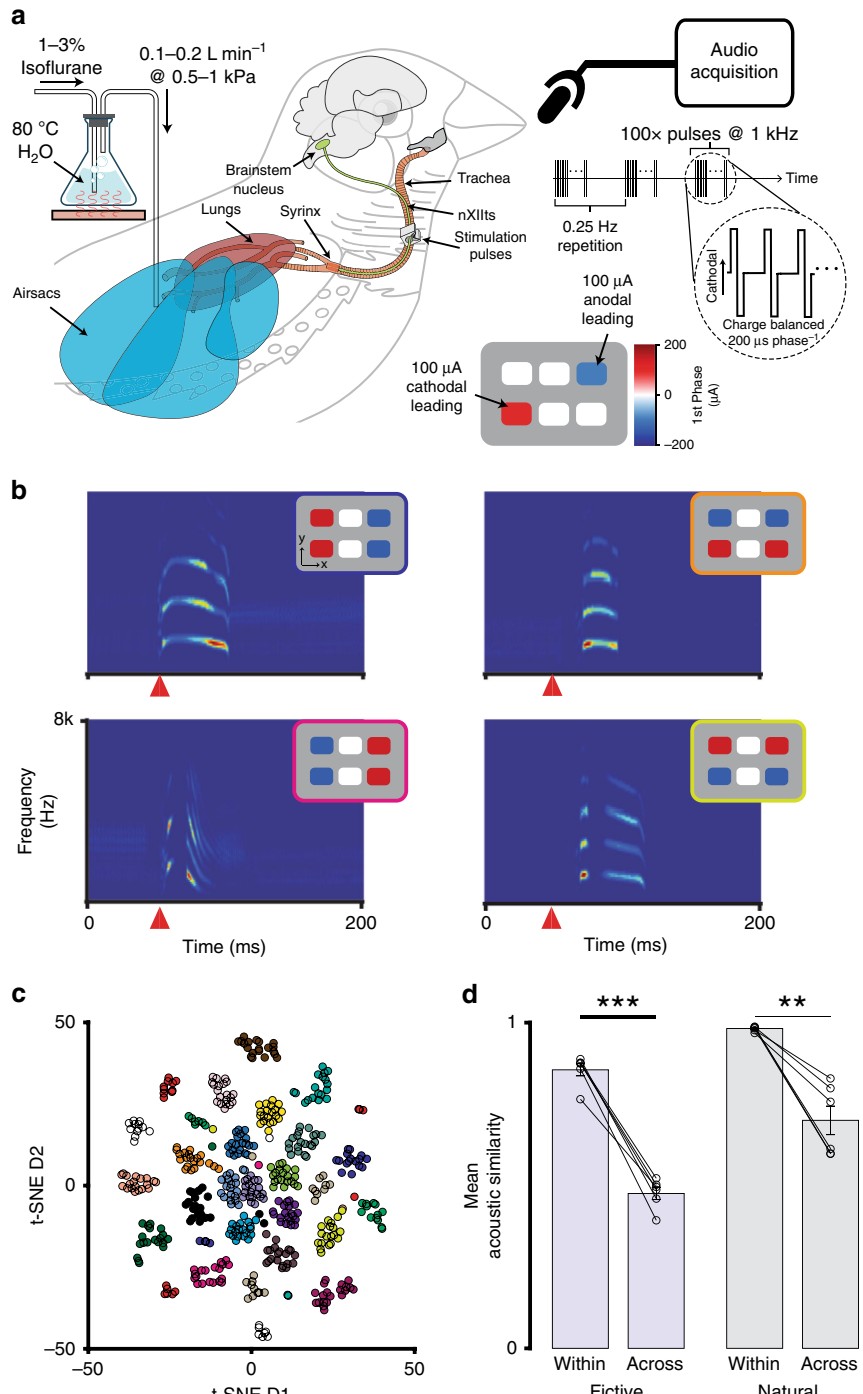

**Fig. 7 Multichannel stimulation in a small volume can achieve functional modulatory specificity. a** In vivo experimental setup for current-steering-evoked fictive singing. The nanoclip was implanted on the right-side nXIIts of an anesthetized adult male zebra finch, and an air cannula was placed in the abdominal air sac (see "Methods"). The respiratory system was pressurized by flowing warmed and humidified (35 C; 80%) isoflurane dissolved in oxygen through the cannula (0.1–0.2 L min$^{-1}$ at 0.5–1.5 kPa). Multichannel stimulation (100 biphasic current pulses at 1 kHz, 200 µs phase$^{-1}$) with unique spatial patterns was applied to the nXIIts with the nanoclip interface, eliciting audible vocalizations that were recorded via a microphone. **b** Representative examples of current-steering-elicited fictive vocalizations. For each pairing, the inset (upper right of each spectrogram) identifies the stimulation pattern that produced the vocalization shown in the spectrogram. Each spectrogram depicts the vocalization from a single trial; the orange triangle marks the onset of stimulation. **c** Two-dimensional t-SNE embedding of high-dimensional representations of fictively produced vocalizations (i.e., spectrograms) from one experiment testing 24 distinct current-steering stimulation patterns. Data points correspond to individual stimulation trials; colors denote current-steering patterns (see Supplementary Fig. 7 for pattern definitions). **d** (Left) Mean acoustic similarity between fictive vocalizations within and across multichannel stimulation patterns for $n = 6$ birds. (Right) Mean acoustic similarity between natural vocalizations within and across song syllable types for $n = 6$ birds. In each, data points show mean across vocalizations within a bird. Bars show mean across all birds in each experiment; error bars indicate sem. Two-tailed, paired $t$-test, within: $P = 3.5 \times 10^{-5}$; across: $P = 0.0016$. **$P < 0.01$; ***$P < 0.001$ Source data are provided as a Source Data File.

To advance bioelectronic therapeutics, the basic science of nerve signaling in relation to physiological function must be expanded. Most existing studies only provide brief snapshots of nerve function—typically at time points close to surgery or later under anesthesia—and may fail to reveal true baseline functioning of nerves[52]. Such intermittent recordings are ill-suited to reliably tracking signals over time, making it difficult to discern how peripheral circuit dynamics are shaped over longer time-scales by developmental, disease, and restorative processes[2,27,53]. Furthermore, the development of future closed-loop bioelectronic therapies that record, decipher, and modulate ongoing activity will require continuous access to high-quality, stable nerve recordings. Chronic neural interfacing with the nanoclip may provide longitudinally stable, bi-directional interfaces in a variety of awake animal studies.

In addition, we demonstrated that the six contacts of the nanoclip can be activated in distinct spatial patterns to differentially modulate the syrinx of the songbird. This multichannel stimulation achieved a high degree of specificity despite the small size of the nanoclip (Figs. 6 and 7). This finding suggests the nanoclip could be used to differentially modulate spatially segregated fascicles in small nerves or, as in the case of the nXIIts, provide tunable modulation for single-fascicle nerves. This prospect also raises the concern that even for small nerves, multi-channel stimulation patterns could have varying effects on the end organ, and a highly stable PNI may be required to produce a consistent effect for a fixed stimulation pattern. Even with bio-mechanical stability, however, full realization of multichannel neuromodulation for precise control of PNS activity will require closed-loop, unsupervised methods for mapping stimulation patterns to functional outputs such that effective control of end state is possible. These technologies are active areas of investigation by the authors and others; taken together, these developments would enable the creation of viable clinical devices for long-term precision interfacing in the periphery.

## Methods

**Fabrication of the nanoclip interface.** The nanoclip interface for chronic peripheral nerve mapping and control used fabrication procedures and design features similar to those of our recent report[26]. Key steps in the fabrication of the device are overviewed in Supplementary Fig. 1 with device dimensions in Supplementary Fig. 2 and as follows: total thin-film device length, 40 mm; neck width, 250 μm electrode array width, 420 μm; electrode array thickness, 12 μm; longitudinal electrode pitch, 80 μm; transverse electrode pitch, 45 μm; gold interconnect line width, 10 μm; number of electrode sites, 6; overall dimensions of nanoclip, $L \times W \times H = 300 \times 300 \times 400$ μm. The key fabrication steps (illustrated with additional details in Supplementary Fig. 1) are as follows: (1) thin-film, multi-site electrode was custom fabricated using polyimide thin-film microfabrication techniques (HD Microsystems PI-2610; Smania, S.R.L.) and subsequently wire bonded to a printed circuit board for connection to test equipment; (2) the electrode was mounted on a thin optical glass substrate (24 × 60 mm, #0-thickness cover glass, Gold Seal) with 150-μm-thick double-sided acrylic tape (Tape #9713, 3M), and a drop (~0.5–2 μL) of liquid acrylic photoresist (IP-Dip, Nanoscribe, GmbH) was deposited over and beneath the recording array; (3) the nanoclip base was printed through the optical glass substrate using a two-photon-polymerization-based, dip-in resonant direct laser writing (rDWL) process;[29] (4) the glass substrate and sample were inverted, and the top half of the nanoclip was printed with the rDWL process; (5) the photoresist was developed by submerging the glass substrate, electrode, and nanoclip in propylene glycol methyl ether acetate (PGMEA; Sigma Aldrich) for 20 min, and the whole device rinsed in methoxy-nonafluorobutane (Novec 7100; 3M) to remove trace PGMEA residue; (6) a three-electrode cell comprising the gold pads as working electrodes, an Ag|AgCl reference electrode, and a large surface area platinum counter electrode was used with an iridium tetrachloride-oxalic acid plating solution to deposit EIROF via cyclic voltammetry (CV), 20–60 cycles at 50 mV/s sweep rate between −0.05 and 0.575 V;[54,55] (7) the electrodes were then rinsed with de-ionized water and characterized by CV and electrochemical impedance spectroscopy in 1x phosphate buffered saline (PBS) (see Supplementary Fig. 3; electrodes were plated to impedance, $Z = 10$–$20$ kΩ at 1 kHz and charge storage capacity, $CSC = 25$–$40$ mC cm$^{-2}$). To confirm the integrity of fabrication, a subset of finished devices was sputter-coated with gold (3 cm distance to gold target; 10 s of sputtering at 0.05 mbar and 20 mA; Sputter Coater 108) and imaged via SEM micrographs (6 mm working distance with the secondary electron sensor,

3 kV accelerating voltage, 10 μm aperture; Zeiss Supra 55VP; see Fig. 1c). To investigate nonneuronal signals recorded in close proximity to devices implanted on the nXIIts, some nanoclips were fabricated with an additional nerve anchor printed on the base so that it could be attached to the nerve without the electrode pads in contact with the nerve. (See Fig. 5a for schematic of "off-nerve" nanoclips.) All mechanical design was performed with Solidworks (Dassault Systèmes). All tested and imaged devices were fabricated using a custom rDLW system[29].

**Mechanical modeling.** All finite element mechanical modeling was performed in Solidworks Simulation (Dassault Systèms) with subsequent analysis using custom MATLAB scripts. The 3D model analyzed was the same as the 3D model from which the nanoclip was fabricated (Supplementary Fig. 2b) with the oval-shaped feet (which link to the nanoclip base through the thin-film electrode) removed (Supplementary Fig. 4a). For all simulations, the bottom edge of the model was fixed in space (indicated by the hatched base in Supplementary Fig. 4c, d). Estimates for IP-Dip mechanical properties listed in Supplementary Fig. 4b were collected from previously published studies[56–59]. Due to the dependence of some IP-Dip properties—and in particular Young's modulus—on the degree of photo-resist polymerization, we report the results of models using both upper- and lower-bound estimates. We assumed linearity of the material until the yield stress. For the simulation, we generated a solid mesh of 10,358 parabolic tetrahedral elements with a mean edge length of $16.5 \pm 0.82$ μm; the maximum element aspect ratio was 4.25 with 99.6% of elements < 3. We simulated the displacement and von Mises stresses of the trap door and hinge in response to graded forces applied uniformly and normal to the exterior surface of the door (Supplementary Fig. 4c, g). In Supplementary Fig. 4f, we report the maximum von Mises stress within the material, regardless of where in the structure (e.g., in which mesh element) it appeared. The full-open position of the door (Supplementary Fig. 4c–g) was defined as the first point of contact between the bottom edge of the door and the interior surface of the nanoclip. For the fatigue analysis, we simulated cyclic stretch of the trap door and hinge with sinusoidal force applied uniformly and normal to the exterior surface of the door at 10 Hz and max amplitude equal to that necessary to fully open doors in the displacement simulation (i.e., 1.25 and 7.5 μN).

**Vertebrate animal subjects.** Adult male zebra finches (*Taeniopygia guttata*; 90+ days after hatch, $n = 37$ birds) were obtained from the Boston University breeding facility and housed on a 13:11 h light/dark cycle in individual sound-attenuating chambers with food and water provided ad libitum. The care and experimental manipulation of the animals were performed in accordance with the guidelines of the National Institutes of Health and were reviewed and approved by the Boston University Institutional Animal Care and Use Committee. Because the behavioral effects of our interventions could not be prespecified prior to the experiments, we chose sample sizes that would allow for identification of outliers and for experimental reproducibility. No animals were excluded from experiments post hoc. The investigators were not blinded to allocation of animals during experiments and outcome assessment.

**Surgical procedures.** All surgical procedures were performed under isoflurane anesthesia (1–4% dissolved in oxygen), with peri-operative analgesia (1% Lidocaine, SC) and anti-inflammatory (1% Meloxicam, IM) regimens. All nerve implants reported in this study targeted the avian hypoglossal cranial nerve, the tracheosyringeal nerve (nXIIts). The nXIIts, which runs along the length of the songbird trachea and terminates at the syrinx, has a diameter of ~150 μm and is composed of both afferent and efferent fibers[26]. At all experiment end points, animals were given an overdose injection of sodium pentobarbital (250 mg kg$^{-1}$ Euthasol, IC).

Acute preparation ($n = 6$ birds): an anesthesia mask was placed over the bird's head and the animal placed in a supine position with a small pillow beneath the neck for support. Feathers were removed from the lower head, neck, and upper chest, and betadine antiseptic solution (5% povidone-iodine) and ethanol (70%) were successively applied to prepare the incision site. A 20–25 mm incision was made at the base of the neck, and the tissue blunt dissected to expose the trachea. Sutures were placed in the skin of the lateral edge of the incision and retracted from the body to expose the implant site. Connective tissue surrounding the nerve was blunt dissected away, and two sections of the nXIIts (each 3–4 mm and ~15 mm apart) were isolated from the trachea. A nanoclip interface was implanted at the rostral location. For the recording experiments in Fig. 3 ($n = 3$ birds), a bipolar silver hook stimulating electrode was placed at the caudal location; for the stimulating experiments reported in Fig. 6 ($n = 3$ birds), a second nanoclip interface was implanted at the caudal location. For all recording experiments, a platinum ground wire (0.003 in dia., Teflon coating; AM-Systems) was sutured to the inside of the skin and away from the neck muscles. Tissue dehydration during the procedure was minimized with generous application of PBS to the nerve and surrounding tissues. At the conclusion of the experiment, the animals were sacrificed and the devices recovered. Individual acute recording and stimulating experiments lasted 2–3 h in total.

Fictive singing preparation ($n = 6$ birds): the bird was prepared for surgery as above, with additional preparation of the skin at the abdomen at the caudal end of the sternum. A single 3–4 mm section of nXIIts was isolated from the trachea and

implanted with a nanoclip interface, and the skin incision closed (around the protruding electrode interconnect) with 2–3 simple interrupted sutures. With the bird in lateral recumbency, a 2 mm silicone cannula was placed in the abdominal air sac, secured to the skin with finger-trap suturing, and the air sac-cannula interface sealed with Kwik-Kast (WPI). The bird was wrapped in elastic nylon mesh and isoflurane delivery switched from the mask to the air sac cannula for the remainder of the experiment. At the conclusion of the experiment, the animals were sacrificed and the devices recovered. Individual fictive singing experiments lasted 3–5 h in total.

Chronic implant for nerve function assessment ($n = 12$ birds): the bird was prepared for surgery and trachea exposed via blunt dissection as in the "acute preparation." Sections of the nXIIts (each 3–4 mm) were isolated bilaterally from the trachea. Each bird received one of the following manipulations: (1) bilateral implant of nanoclip nerve interface ($n = 3$ birds); (2) bilateral sham implant ($n = 3$ birds); (3) bilateral nerve crush (transient ~1 N force applied to nXIIts with 2.5 mm wide flat forceps; $n = 3$ birds). Anti-inflammatory splash block (~250 μL 1% Meloxicam) was applied directly to intervention site, and incisions were closed with sutures. All birds exhibited normal rates of singing within 1 day of surgery. For intact controls ($n = 3$ birds), no surgical, anesthetic, or analgesic measures were taken.

Chronic implant for nerve recording ($n = 7$ birds): the bird was prepared for surgery as in the "acute preparation" and placed in a stereotax. A sagittal incision was made along the top of the head and the tissue retracted. Four to six stainless steel anchor pins (26002–10, Fine Science Tools) were threaded between layers of the skull, and a head cap was made from dental acrylic. The device connector was secured to the head cap with additional dental acrylic, and the nanoclip and a platinum reference wire (0.003 in dia., Teflon coated; AM-Systems) were trocared beneath the skin to the neck. The animal was then removed from the stereotax and placed in supine position with neck support. A 10–15 mm incision was made at the base of the neck, and the trachea and nXIIts were isolated as described above. The nanoclip interface was implanted on the nerve, and the reference wire secured to the underside of the skin. In $n = 2$ birds, a second interface with an off-nerve nanoclip (see Fig. 5a) was implanted on the same nXIIts nerve ~5 mm caudal to the first device. Anti-inflammatory splash block (~250 μL 1% Meloxicam) was applied directly to the implant site, and incisions were closed with sutures. All birds exhibited normal rates of singing within 1 day of surgery; birds acclimated to tethering and resumed singing within 1–2 days.

**In vivo electrophysiology**. All experiments were implemented and controlled using custom LabVIEW (National Instruments) and MATLAB (MathWorks) software applications.

Acute electrophysiology: acute electrophysiological data were recorded on the right-side nXIIts using nanoclip interfaces with a RZ5 BioAmp Processor and an RA16PA Medusa Preamplifier (Tucker-Davis Technologies). Neural signals were digitized at 24.4 kHz and 16-bit depth and were Bessel bandpass filtered (1 Hz–10 kHz, zero-phase). Stimulation currents were delivered—through either bipolar silver hook electrodes or a nanoclip interface—using a PlexStim programmable stimulator (Plexon). For all acute electrophysiology experiments, current pulses were biphasic, 200 μs phase$^{-1}$ in duration, delivered at 1 Hz, and varied in amplitudes from −200 to 200 μA. By convention, positive current amplitudes are cathodic; negative amplitudes anodic.

Fictive singing: 1–1.5% isoflurane in dissolved oxygen, warmed and humidified to near-physiological levels (35 C, 80% humidity) by bubbler cascade to minimize respiratory tissue damage and prolong work time, was delivered by an air sac cannula (Fig. 7a). Flow rates were adjusted to maintain a stable anesthesia plane while eliminating broadband noise from forced gas flow over the passive syringeal labia (typically 100–200 mL min$^{-1}$ at 0.5–1.5 kPa). Fictive vocalizations were recorded with an omni-directional condenser microphone (AT-803, Audiotechnica) placed 5–10 cm from the bird's open beak, amplified and bandpass filtered (10x, 0.1–8 kHz; Ultragain Pro MIC2200, Behringer), and digitized by data acquisition boards (PCIe-6212, National Instruments) and LabVIEW software at 44.15 kHz and 16-bit depth. Current-controlled stimulation was delivered to the right-side nXIIts through a nanoclip interface using a programmable stimulator (PlexStim, Plexon) with a custom MATLAB interface. 1 kHz bursts of 100 biphasic pulses, 200 μs phase$^{-1}$ in duration, delivered at 0.25 Hz, and varying in amplitudes from −200 to 200 μA were delivered at each active stimulation site (of up to six in total). The current-steering parameters used here consisted of spatially distinct multichannel stimulation patterns (all current-steering patterns tested are listed in Supplementary Fig. 7).

Chronic electrophysiology: all birds were recorded continuously using song-triggered LABVIEW software as above, generating a complete record of vocalizations and nerve activity for the experiment. Song-detection thresholds were set to detect periods in which power in the 2500–8000 Hz band (corresponding to song) exceeded 10–50 times the power in the 50–250 Hz band (corresponding to low frequency background noise); recordings continued 1.5 s after the cessation of song. Neural recordings were simultaneously acquired with an RHD 2000 system with a 16-channel unipolar input headstage (Intan Technologies), amplified, and bandpass filtered (0.3–15 kHz). Singing-related nerve activity was recorded from up to six sites on the right-side nXIIts in $n = 7$ birds. In $n = 3$ birds, recordings were made for >30 days before implant failure; these animals are reported in Fig. 4.

In $n = 2$ birds, chronic recordings were made for 17 and 19 days before implant failure; postmortem inspection found evidence of fatigue-related fracture of the polyimide electrode along the interconnect. In an additional $n = 2$ birds, nonneuronal signals from a second off-nerve interface were recorded simultaneous with neuronal signals from an on-nerve interface (Fig. 5 and Supplementary Fig. 6). Across all chronically recorded birds, we found high correlation between simultaneous nerve recordings made at adjacent recording sites; we report on data collected at the most stable recording site in each bird, though we note that the trends were similar across all channels.

**Data analysis**. All song and electrophysiology data analysis was performed off-line using MATLAB.

Stimulation evoked responses: we sampled activity ~5 ms before and up to 25 ms after stimulation onset and used the onset of the stimulation artifact (Figs. 3b, d and 6b, c at 0 ms) to temporally align individual trial responses. Absolute response amplitudes were observed and quantified in a stimulation response window of 0.75–4 ms after stimulation onset—a latency consistent with estimated nerve conduction velocities for 4–6 μm diameter myelinated axons (i.e., 4–24 m s$^{-1}$)[11]. SNR was calculated from mean recordings ($n = 20$ trials) made in the stimulation response window (i.e., signal) and 10.75–14 ms after stimulation (i.e., noise). We considered an evoked response to be detected if the SNR within the signal response window exceeded a 90% confidence interval calculated by bootstrap (i.e., resampling with replacement the signal and noise intervals over $n = 10,000$ trials). Figures 3b and 6b show individual stimulation trials from single experimental sessions. Data points in Fig. 3c show mean response over $n = 20$ trials for each bird; symbols identify individual birds. Figure 3d shows the mean (solid line) and standard error (shaded region) across trials. Figure 3e shows the mean (bar) and standard deviation (error bars) across animals; symbols identify individual animals.

Syllable segmentation and annotation: raw audio recordings were segmented into syllables using amplitude thresholding and annotated using semi-automated classification methods. Briefly, spectrograms were calculated for all prospective syllables, and a neural network (5000 input layer, 100 hidden layer, 3–10 output layer neurons) was trained to identify syllable types using a test data set created manually by visual inspection of song spectrograms. Accuracy of the automated annotation was verified by visual inspection of a subset of syllable spectrograms.

Syllable acoustic feature quantification and acoustic similarity: both natural song syllables and fictive vocalizations were characterized by their pitch, frequency modulation, amplitude modulation, Wiener entropy, and sound envelope—robust acoustic features that are tightly controlled in adult zebra finch song. Each feature was calculated for 10 ms time windows, advancing in steps of 1 ms, such that an estimate was computed for every millisecond. Acoustic similarity between vocal elements was calculated by pairwise millisecond-by-millisecond comparisons of acoustic features of identified vocalizations in five-dimensional space. The median Euclidean distance between points was converted into a $P$ value based on the cumulative distribution of distances calculated between 20 unrelated songs. Thus, on this scale a similarity score of 1 means that syllables are acoustically identical, while a score of 0 indicates that the sounds are as different as two unrelated syllables.

Assessing nerve function via acoustic similarity: to assess nerve function following bilateral nanoclip implant (or sham, crush, and intact controls), we calculated the acoustic similarity of songs produced before implant with those produced following manipulation. This was quantified for each experimental group as the mean across syllables within a bird (100 renditions each of 3–6 unique syllable types) and then the mean over birds within a treatment group ($n = 3$ birds per group). Figure 2b shows this as the mean (solid line) and standard deviation (error bars) for each treatment group, normalized to the group mean of pre-manipulation acoustic similarity. Figure 2c summarizes post-implant changes in acoustic similarity as the mean (bars) and standard deviation (error bars) changes from pre-manipulation baseline on days 1 and 8; circles identify individual animals.

Quantifying specificity of stimulation via acoustic similarity: to quantify the specificity of current-steering stimulation, we compared the acoustic similarity of fictive vocalizations elicited within the same stimulation pattern with those produced across different stimulation patterns. (See Supplementary Fig. 7 for stimulation pattern definitions.) This is calculated as the mean over vocalizations elicited by stimulation type (i.e., within or across stimulation patterns; ~20 trials of up to 24 patterns) within a bird and then the mean over birds for each stimulation type ($n = 6$ birds). For comparison, the same analysis was performed on naturally produced vocalizations from unmanipulated birds ($n = 6$). Figure 7d shows this as the mean (bar) and standard deviation (error bars) for each stimulation type and experimental group (i.e., fictive or natural vocalizations); circles identify individual animals.

t-SNE embedding of fictive vocalizations: fictive vocalizations produced from ~20 trials of up to 24 stimulation patterns were included in this visualization. Audio recordings of each fictive vocalization, starting at the onset of stimulation and lasting for 200 ms, were converted into spectrograms (5 ms window, 1 ms advance, 512-point nfft). Spectrogram rows corresponding to frequencies above 8 kHz were discarded, and the remaining matrices (200 timesteps by 91 frequency bins for each vocalization) were transformed into 18,200-dimension vectors. The data set dimensionality was reduced to 50 with principal components analysis, and these data were subsequently embedded in two-dimensional space using t-SNE

with distances calculated in Euclidean space and a perplexity of 35. Figure 7c shows this embedding in a representative bird.

Alignment of the electrophysiology recordings to song: a dynamic time warping algorithm was used to align individual song motifs to a common template[60]. The warping path derived from this alignment was then applied to the corresponding common mode subtracted and bandpass filtered voltage recordings (0.3–6 kHz, zero-phase, two-pole Butterworth) with no premotor time shifting. Inspired by refs. [17,60], the aligned signals were squared (to calculate signal envelope) and smoothed (Fig. 4b: 20 ms boxcar window; all other analyses: 5 ms boxcar window, 1 ms advance).

Neural activity envelope correlation: the trial-by-trial stability of singing-related nerve activity was calculated as the correlation between the song-aligned neuronal signal envelope on the 1st day of recording with those at later time points. The running correlation (Fig. 4d) shows Pearson's correlation between the mean activity envelope of motifs on the 1st day of recording and the mean envelope in a sliding window (width: 25; advance: 1). The data points in Fig. 4e denote the mean correlation between the mean of all signal envelopes produced on day 1 and the mean of signal envelopes in a sliding window (width: 25; advance: 1) produced in a day; symbols identify individual birds.

Neural activity Vpp: the trial-by-trial Vpp of singing-related nerve activity was calculated as the difference of the maximum and minimum voltage recorded for each song motif. The data points in Fig. 4e denote the mean Vpp over all trials produced in a day; symbols identify individual birds.

Neural activity SNR: the trial-by-trial SNR of singing-related nerve activity was calculated using the formula $10 \times \log_{10}(\text{RMS}_\text{S}/\text{RMS}_\text{N})$, where $\text{RMS}_\text{S}$ and $\text{RMS}_\text{N}$ are the root mean square of the signals corresponding to singing and non-singing, respectively. Because our song-triggered recording system captures audio and electrophysiology data after singing ends, we used 500 ms segments of this vocalization-free recording as the in vivo "noise" floor. A subset of these "noise" recordings was visually inspected to confirm the absence of vocalizations or other artifacts. The data points in Fig. 4e denote the mean SNR over all trials produced in a day; symbols identify individual birds.

Neural activity event rate: the trial-by-trial event rate of singing-related nerve activity was calculated as the number of envelope threshold crossings per unit time. A unique threshold was calculated for each motif at 5 standard deviations over the mean during singing; duration of unwarped song was used to calculate rates. The data points in Fig. 4e denote the mean event rate over all trials produced in a day; symbols identify birds.

Comparison of on-nerve and off-nerve recordings: voltage signals recorded simultaneously during singing from on-nerve and off-nerve nanoclips were common mode subtracted (calculated independently for each device) and processed as described above. Trial-by-trial SNR for on- and off-nerve signals was calculated as described above; the data points in Fig. 5c and Supplementary Fig. 6d denote the SNR for on-nerve and off-nerve recordings for individual trials. Trial-by-trial correlation between on- and off-nerve recordings was calculated as the Pearson's correlation of the signal envelopes; the data points in Figs. 5d and S6c denote the correlation between on-nerve and off-nerve recordings for individual trials. Data points in Fig. 5e show mean (marker) and standard deviation (error bars) for all trials in each bird ($n = 154$ and 57); symbols identify individual birds.

**Statistical analysis.** All statistics on data pooled across animals is reported in the main text as mean ± SD and depicted in figure error bars as mean ± SD, unless otherwise noted. Figure starring schema: $*P < 0.05$, $**P < 0.01$, $***P < 0.001$. Where appropriate, distributions passed tests for normality (Kolmogorov-Smirnov), equal variance (Levene), and/or sphericity (Mauchly), unless otherwise noted. Multiple comparison corrected tests were used where justified. Statistical tests for specific experiments were performed as described below:

Figure 2c: changes in the acoustic structure of baseline and post-manipulation song following bilateral nanoclip implant ($n = 3$ birds), nerve crush ($n = 3$ birds), sham implant ($n = 3$ birds), and intact controls ($n = 3$ birds). A two-tailed, paired $t$-test revealed that the changes in acoustic similarity to baseline following nerve crush were significantly different from 0 on day 1 ($P = 0.02$) and day 8 ($P = 0.016$). Days 1 and 8 changes in acoustic similarity were not significantly different from 0 for nanoclip implants ($P = 0.073$ and 0.417), sham implant ($P = 0.53$ and 0.2786), and intact controls ($P = 0.70$ and 0.73). In addition, two-tailed, unpaired $t$-tests showed significant differences between nerve crush and intact controls at all post-surgery time points (i.e., through the 8th day: $P < 0.01$); no significant differences were found between nanoclip implants and intact controls at any time point ($P > 0.22$).

Figure 3e: comparison of stimulation evoked response amplitudes before and after lidocaine/saline application in $n = 3$ birds. Mauchley's test indicated a violation of sphericity ($W = 0$, $P = 0$), and a Huynh–Feldt degree of freedom correction was applied. Subsequent repeated-measures ANOVA revealed significant differences between the treatments ($F_{(1.11,2.21)} = 189.02$, $P = 0.003$). Post hoc comparisons using Dunnett's test showed significant differences between saline (control) and lidocaine application ($P = 6 \times 10^{-6}$); no other condition significantly differed from control ($P > 0.15$).

Figure 4e: quantification of the stability of nXIIts dynamics, Vpp, SNR, and event rate over 30 days of continuous recording in $n = 3$ birds; no statistics were found to change significantly from day 1 over the duration of the experiment. A two-tailed, paired $t$-test revealed no significant differences in correlation between

recording day 1 and days 10, 20, and 30 ($P = 0.36$, 0.089, and 0.179, respectively). A two-tailed, paired $t$-test revealed no significant differences in the Vpp of singing-related activity on day 1 and days 10, 20, and 30 ($P = 0.831$, 0.739, and 0.156, respectively). A two-tailed, paired $t$-test revealed no significant differences in the SNR of singing-related activity on day 1 and days 10, 20, and 30 ($P = 0.504$, 0.208, and 0.311, respectively). A two-tailed, paired $t$-test revealed no significant differences in the event rate of singing-related activity on day 1 and days 10, 20, and 30 ($P = 0.714$, 0.478, and 0.147, respectively).

Figure 5e: comparison of simultaneous on-nerve and off-nerve recordings in $n = 2$ birds. A two-tailed, paired $t$-test revealed no significant differences between mean on-nerve and off-nerve SNRs ($P = 0.2567$). Two-tailed, paired $t$-tests also showed that the mean correlation between on- and off-nerve recordings was not significantly different from 0 ($P = 0.127$).

Figure 7d: mean acoustic similarity of vocalizations within and across multichannel stimulation patterns ($n = 6$ birds) and natural vocalizations ($n = 6$ birds). A two-tailed, paired $t$-test showed that fictive vocalizations were less similar across different stimulation patterns than within repeated application of the same pattern ($P = 3.5 \times 10^{-5}$). A two-tailed, paired $t$-test showed that naturally produced song syllables were less similar across identified syllables types than within the same syllable type ($P = 0.0016$).

**Reporting summary**. Further information on research design is available in the Nature Research Reporting Summary linked to this article.

## Data availability

Additional data supporting the findings of this study are available at https://github.com/timotchy/Otchy-et-al-2020. All other data are available from the corresponding author upon request. Source data are provided with this paper.

## Code availability

All custom code is accessible in an online repository at https://github.com/timotchy/Otchy-et-al-2020.

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

## Acknowledgements

The authors thank Silvia Bossi for fabricating the thin-film arrays at Smania, S.R.L. We also thank Ian Davison, Benjamin Scott, Jeff Gavornik, Alberto Cruz-Martin, and members of the Gardner Lab for helpful comments on drafts of this paper. This research was supported by the NIH (R01NS089679 and R01NS104925) and a sponsored research agreement with GlaxoSmithKline.

## Author contributions

T.M.O., C.M., A.E.W., and T.J.G. conceptualized the project, designed the device, and developed the fabrication method with input from B.J.H. and D.J.C. T.M.O., B.L., and

K.G. performed surgical procedures and in vivo experiments. T.M.O., K.G., J.G., and
V.N. analyzed experimental data. L.D. performed SEM sample preparation and imaging.
D.S. performed immunohistology, biological sample imaging, and animal husbandry.
T.M.O. drafted the paper with input from all other authors.

## Competing interests

T.J.G. is a scientific advisor to Abbott Neuromodulation; B.J.H. and D.J.C were employed
by GlaxoSmithKline. T.M.O., C.M., B.J.H., D.J.C., A.E.W., and T.J.G. are co-inventors of
the nanoclip as described in a pending patent filing (US Patent PR66142P, US Serial No.
62/367,975). The remaining authors declare no competing interests.

## Additional information

020-18032-4.

**Peer review information** *Nature Communications* thanks Cristin Welle and the other,
anonymous, reviewer(s) for their contribution to the peer review of this work. Peer
reviewer reports are available.

**Publisher's note** Springer Nature remains neutral with regard to jurisdictional claims in
published maps and institutional affiliations.

