## [Peer Review File · Nature Communications]

Reviewers' comments:

Reviewer #1 (Remarks to the Author):

This paper presents a neural interface device for the interfacing of small peripheral nerves, which is a difficult problem. The overall goal of this work is the chronic recording and modulation of peripheral nerves for therapeutic control of target organs. The researchers present an extraneural device made of the unique combination of thin film technology and a 3D printed nerve holder with precise geometry. The study showed in vivo implantation of this device for both acute and chronic recording and stimulation of a small nerve controlling the songbird syrinx. They characterize the stability of the device through recorded neural activity and consistency of songbird vocalizations over time. They also present an in vivo acute experimental set up where stimulation from the nanoclip produces artificial vocalization as a method of exhibiting current steering patterns. Overall, this appears to be a useful tool of broad applicability, and is particularly remarkable in the context of a very small 150 μm nerve. However, several major and minor concerns must be addressed prior to publication.

Major criticisms

No comparison is presented to any other device, such that one can judge against the state of the art. At minimum historical or literature controls should be quantitatively evaluated.

Quantification of the histological response after chronic implant for all 4 animals should be shown if it will be discussed. At present, this is qualitatively described, with only a photo of one animal shown, whereas this should be easy to measure from these images. If some data has been lost, that should be noted, and it should be clear what the distances and n is for the histological claim.

While the unit data presented appears to be impressive, this is a strong claim that should be rigorously evaluated. The best evidence is if the song is tightly linked to the putative unit data, as it appears to be in Figure 4c and 4d. However, in 4b at the top, there is no clear relationship between the ABC motifs and the putative unit activity, which seems inconsistent, and this is also inconsistent with 4D. It is also unclear which birds and how many are used throughout this figure. The authors should also provide enough information to understand whether most data is from 1 exemplary bird out of the 3 by quantifying the μV amplitudes (i.e. not normalized as shown) for all 3 birds.

EMG invasion would seem to be a major concern, given the nearby muscles of the trachea and syrinx. This could produce the apparent relationship to the song as shown, despite no signals being recorded from the nerve. Nerve cuffs often explicitly incorporate shielding because this is such a common issue. The authors should confirm that there is no nearby muscle that could produce signals that could be misinterpreted as nerve activity, or else confirm this with simultaneous recording.

Minor criticisms

Editing for wordiness and run-on sentences is needed throughout the document.

Other thin film nerve cuffs should be cited and discussed, such as TEENI (Desai et al 2017), tf-LIFE, TIME, and SELINE (Wurth et al.)

P-values are needed with all statements of significance in the results section.

What was the stimulation threshold current used to produce the minimum evoked response in Figure 3?

Correlation and mean amplitude plots in Figure 4D are unclear in the text and caption.

Can you describe the forces exerted on the nerve when pressed against the door via comparison to a similar technique? Similarly, fatigue characterization of the trap doors would be of interest. It should also be discussed whether the device can be removed from the nerve and repositioned during surgery.

It is unclear if the nano clip can move along the length of the nerve. Is there an anchoring method used to stabilize the clip in a certain place?

Five animals were recorded until day 30, but 3 animals were recorded past the first month. For completeness, what was the reason for 2 animals not being recorded past the first month?

It is important to see the examples of fictive vocalizations in Figure 5e, but the figure is extremely small and hard to see.

Is the reason behind the lower stimulation threshold for the nano clip than the bipolar silver hook the better insulation provided by the nano clip?

Reviewer #2 (Remarks to the Author):

This paper presents a novel microscale implantable device capable of recording from and stimulating small nerves. The engineering feats are coupled with detailed electrophysiological and behavioral analysis in the songbird vocal system. While a simpler version of these devices has been published earlier (Lissandrello et al., 2017), this device is novel as it incorporates a microfabricated, multi-channel thin film electrode substrate, which allows for more precise stimulation and recording than the previous version. In addition, this paper presents chronic behavioral and electrophysiological data, which has not before been published.

In general, the data presented in this paper constitute a ground-breaking advance in small nerve recording and modulation, and provides a tool for more detailed study of nerve signals and function than currently exists. However, figures tend to lack detail, particularly in the realm of data analysis, and should be improved prior to publication. A list of major concerns is presented below:

1. Figure 2 – Histology is mentioned in the text, but no histological images or quantitative assessment is provided. Please provide evidence of nerve integrity with implantation of this device, both qualitative and quantitative.
2. Figure 2B- Please provide quantitative analysis of the difference in spectral similarity between nerve crush, sham and Nanoclip implant from non-injured baseline similarity.
3. Figure 4 presents evidence for the stability of nerve recordings across 30 days. However, the metric used to quantify these recordings, a smoothed activity envelope, is atypical for neural recordings. While not improper, a more conventional metric would be more meaningful for future neurophysiological applications of this technology. For instance, Zanos et al. recently presented vagal nerve recordings that used wavelet decomposition to identify unique compound action potential signals (CAPS), and then reported event rate, inter-cap-interval histograms and signal-to-noise ratio for each CAP, metrics that are easily comparable to single unit recordings from the brain (Zanos et al., 2018). Given the author's background in neurophysiological recordings in the songbird (ie. Ölveczky et al., 2011), it is surprising that a more detailed analysis of neurophysiological data was not applied. I would recommend the authors add additional neurophysiological analysis to demonstrate that this technology can be meaningful to the songbird field, and more broadly.

4. The longevity data in Figure 4 looked very promising and exciting. Again, I would have preferred to see this data quantified more extensively – ie. Signal-to-noise, event rate. Predictive value for syllable shape would be amazing, although not required.

5. Figure 5 shows specificity of stimulation across different contact sites. The variability is quantified by tSNE plot. While tSNE offers the advantages of being an unsupervised non-linear visualization technique, it can be prone to misinterpretation, and can change shape with differing perplexity and step values. In particular, distances between and within clusters can change dramatically with different iterations. Replication of the results in Fig. 5F,G using more conventional syllable variability metrics are necessary to support tSNE measurements, and the overall conclusion regarding stimulation specificity.

Minor points:

Fig. 4D – please provide a color scale for the color map.

Fig. 4D – the text box overlaps the figure labels.

Fig. 4F – does this refer to the signal envelope, or the raw (processed?) signal?

Lissandrello CA, Gillis WF, Shen J, Pearre BW, Vitale F, Pasquali M, Holinski BJ, Chew DJ, White AE, Gardner TJ (2017) A micro-scale printable nanoclip for electrical stimulation and recording in small nerves. *J Neural Eng* 14:036006.

Ölveczky BP, Otchy TM, Goldberg JH, Aronov D, Fee MS (2011) Changes in the neural control of a complex motor sequence during learning. *J Neurophysiol* 106:386–397.

Zanos TP, Silverman HA, Levy T, Tsaava T, Battinelli E, Lorraine PW, Ashe JM, Chavan SS, Tracey KJ, Bouton CE (2018) Identification of cytokine-specific sensory neural signals by decoding murine vagus nerve activity. *Proc Natl Acad Sci U S A* 115:E4843–E4852.

Cristin Welle

Reviewer #3 (Remarks to the Author):

This exciting manuscript presents a new device/system for recording and manipulating the activity of small peripheral nerves, *in vivo*, during behavior and over periods as long as 1 month. Small-scale engineering and experimental neurobiology are brought together to develop small, flexible, electrode arrays that are secured to the surface of an intact nerve such that compound action potentials can be recorded and nerve fibers can be stimulated in small, freely behaving animals during behavior. The function of this new interface is demonstrated with multiple experiments. Because development of such systems are important for understanding how neural signals in the brain are transformed by the peripheral nervous system into muscle activity, the results of this study will be impactful for experimental and clinical neuroscience. The manuscript is well written; the system components, experiments, data and analyses are clearly described. Below, I suggest improvements to the manuscript that would further demonstrate the system's safety and broad-scale usefulness.

Major

1. The choice of the zebra finch for testing the system is wise because the anatomy and physiology of the PNS involved in singing are well described and the motor production of song is highly stereotyped. Here, the tracheosyringeal nerve, which innervates the muscles of the syrinx, and singing behavior are studied to test the usefulness of the system. First, the authors show that they reliably obtained interpretable nerve recordings over time (indicating little to no nerve damage) and across animals (indicating precise and consistent placement of electrode contacts). Second they show that the system can be used to evoke nerve activity via electrical stimulation that is

specific to the spatial pattern of stimulating electrode contacts, and that results in vocal output. While these demonstrations are strong, they could be improved without collecting more data. First, the results of the histological analyses of nerve segments that were housed in the clip for weeks should be added. If the quantified analyses of control and clipped nerve segments shows that the clip does not damage the nerve fibers even after a month of implantation, then the success of the clip design will be strongly supported. Second, the Results should include information on the estimated number and spatial extent of fibers that are stimulated by each electrode contact and how that relates to the experimental observations of different activity patterns and vocalizations when different spatial patterns of stimulation are used. The addition of this information will strengthen the argument that the system can be used to manipulate nerve activity in a precise manner.

2. The broad scale usefulness of a new approach/system is clearest when the system is tested in more than one experimental context, here, a nerve. With the goals in the Introduction in mind, the paper would be stronger if clipping, recording and stimulating were demonstrated in a second nerve. A candidate is the 8th nerve, which is accessible and reasonably well studied in birds. The addition of these data would show the generalizability of implant placement without functional damage, accurate recording of nerve activity and the effectiveness of stimulation.

Minor

1. Add detection threshold to Figure 4c.
2. Add one high resolution plot of the smoothed activity for at least 1 ABC sequence to Figure 4d. The data shown as a heatmap over recording days is great, but an intermediate step between panels c and d is needed.
3. If I understand the Figure 5 experiments, panels a – c and d – g describe completely different experiments. The top panels show experiments with 2 clips, one recording and one stimulating. The bottom panels show experiments using only a stimulating clip, and vocal production with stimulation of the nerve and air pressure in the respiratory system. If this is correct, then Figure 5 should be revised into 2 figures, each showing one of these experiments.
4. The full paragraph on the second page of the Discussion includes sentences, phrases and words that systems neuroscience readers with not understand without explanation. Examples are “present elastic moduli substantially above the kilopascal range of the host tissue” and “despite the Young’s modulus of their bulk material being within the gigapascal range.” Please add explanation.
5. The Methods section includes a subsection on the quantification of 5 syllable features. But these features are not in the Results.
6. Typo in the Methods “was calculated was calculated”
7. Typo in the Methods “20 unrelated.”
8. Add page numbers or line numbers to the ms.

Response to the reviewers' comments

We thank the reviewers for attesting to the novelty, utility, and import of the nanoclip and for their insightful and constructive comments. In response, we have added new experiments (simultaneous on- and off-nerve chronic recordings), re-analyses of data reported in the original submission, mechanical simulations of the nanoclip, new main and supplementary figures, and expanded discussion of results. In addition, we have carefully revised the text to clarify issues reviewers raised and believe the manuscript has significantly improved as a result. We hope the reviewers agree.

Reviewer #1

This paper presents a neural interface device for the interfacing of small peripheral nerves, which is a difficult problem. The overall goal of this work is the chronic recording and modulation of peripheral nerves for therapeutic control of target organs. The researchers present a extraneural device made of the unique combination of thin film technology and a 3D printed nerve holder with precise geometry. The study showed in vivo implantation of this device for both acute and chronic recording and stimulation of a small nerve controlling the songbird syrinx. They characterize the stability of the device through recorded neural activity and consistency of songbird vocalizations over time. They also present an in vivo acute experimental set up where stimulation from the nanoclip produces artificial vocalization as a method of exhibiting current steering patterns. Overall, this appears to be a useful tool of broad applicability, and is it particularly remarkable in the context of a very small 150 um nerve. However, several major and minor concerns must be addressed prior to publication.

Major criticisms

1. No comparison is presented to any other device, such that one can judge against the state of the art. At minimum historical or literature controls should be quantitatively evaluated.

We agree that a comparison of nanoclip performance to other neural interfaces would strengthen the manuscript and aid in demonstrating the relative merits of our device. In the revised manuscript, we include an expanded discussion of peripheral nerve interfaces with emphasis on extra-neural approaches, of which the nanoclip is one recent addition. In addition, we have added Table 1 which provides several representative examples of extra-neural interface designs, application, size, and quantitative performance metrics. For our songbird test model, we could not find any similar alternative devices on this scale to test experimentally.

2. Quantification of the histological response after chronic implant for all 4 animals should be shown if it will be discussed. At present, this is qualitatively described, with only a photo of one animal shown, whereas this should be easy to measure from these images. If some data has been lost, that should be noted, and it should be clear what the distances and n is for the histological claim.

We thank the reviewer for this request, and we agree that additional data and quantitative analyses are required to support claims of reduced inflammatory response and fibrotic penetration. We provided a qualitative description of the histology response as low sectioning yield (due to frequent fracture of the acrylic nanoclip) limited thorough quantitative analysis. In addition, the songbird model used here lacks well-defined immuno-histological markers for inflammation and fibrotic tissue, making it poorly suited to the histological analysis warranted.

Prompted in part by these requests, we are planning new experiments to address the long term bio-integration and safety of the nanoclip for the implanted nerve and intrafascicular blood supply in a clinically relevant model (i.e., rats). These rat studies will take advantage of established protocols for quantifying implantation-related inflammation and fibrosis. Though important, these additional experiments and analyses are beyond the scope of the current manuscript and will be reserved for a future report. In this revised manuscript, we have removed references to histology, reactive tissue responses, and fibrotic penetration into the nanoclip. We now make only the claim “In all chronically implanted animals for which necropsies were performed ($n = 5$ birds), the nanoclips were found securely latched over the nerve up to 11 months after implant, suggesting the potential for long-term viability of the preparation.”

3. While the unit data presented appears to be impressive, this is a strong claim that should be rigorously evaluated. The best evidence is if the song is tightly linked to the putative unit data, as it appears to be in Figure 4c and 4d. However, in 4b at the top, there is no clear relationship between the ABC motifs and the putative unit activity, which seems inconsistent, and this is also inconsistent with 4D. It also also unclear which birds and how many are used throughout this figure. The authors should also provide enough information to understand whether most data is from 1 exemplary bird out of the 3 by quantifying the uV amplitudes (i.e. not normalized as shown) for all 3 birds.

We completely agree with the reviewer and regret that the presentation in Figure 4 was not as clear as it could have been. With respect to the relationship between the ABC motifs and the putative unit activity in Figure 4b, we believe that a combination of resolution loss during export from MATLAB and our use of a heavier weight line in a dense recording obscured the finer structure of the activity. In this revision, we re-plotted the panel to better highlight the modulation of activity during singing. Specifically, we believe this better illustrates the increase in activity corresponding to the syllable A to syllable B transition that is a prominent feature of the envelope plots at the bottom of Figure 4b, as well as in 4c and 4d. We note also that the newly added Figure 5 includes an additional example of activity during singing that we believe also demonstrates the reliable relationship between behavior and recorded nXIIIts signals.

In addition, we have revised the Figure 4 caption to clarify that the examples of nXIIIts recordings in subpanels b-d are from a single representative animal. Subpanel e reports summary data from the three animals that were recorded for more than 30 days, with the marker shape consistently identifying each animal. Finally, we agree with the reviewer that the normalized mean amplitude metric was opaque and challenging to interpret – both on its own and in comparison to existing recording technologies. In this revision, we have replaced this in Figure 4d,e with un-normalized measures of the signal-to-noise ratio and event rate of singing-related activity. We believe these changes, which were also suggested by

another reviewer, are both responsive to your concern and more interpretable for readers.

4. EMG invasion would seem to be a major concern, given the nearby muscles of the trachea and syrinx. This could produce the apparent relationship to the song as shown, despite no signals being recorded from the nerve. Nerve cuffs often explicitly incorporate shielding because this is such a common issue. The authors should confirm that there is no nearby muscle that could produce signals that could be misinterpreted as nerve activity, or else confirm this with simultaneous recording.

We could not agree more with the reviewer that artifact invasion of the recordings is a possible explanation that must be excluded. We have now performed new experiments to address directly the neuronal-origin of nanoclip-recorded signals, and we include new figures (Figures 5, S6) and text reporting these results.

We made simultaneous recordings from nanoclip interfaces with electrodes in contact with (i.e., 'on-nerve') and adjacent to (i.e., 'off-nerve') the nXIIIts. For off-nerve recordings, we printed a second nerve anchor on the underside of a nanoclip such that the interface could be implanted on the nXIIIts while positioning the electrode $\sim 150 \mu\text{m}$ from the nerve surface with the electrodes facing away from the nerve and exposed to the extra-neural environment (see Figure 5a for schematics of this preparation). The on-nerve and off-nerve interfaces were implanted on the same nerve $\sim 5 \text{ mm}$ apart; given the anatomy, the two nanoclips were approximately equidistant from the proximal muscles. Thus, we reasoned that each interface would be subject to the same biomechanical and electrochemical (i.e., non-neuronal) artifact sources, but only the 'on-nerve' recordings would reflect the additional contribution of nXIIIts neuronal activity at the surface of the electrode pads. In $n = 2$ birds, we recorded with on-nerve and off-nerve electrodes voltage signals that were temporally correlated with singing (see Figure 5b, top). As in the original single-nanoclip recording experiments, the on-nerve recordings showed well-defined voltage amplitude fluctuations in the $500 \mu\text{V}$ range; however, the off-nerve recordings were poorly defined and of consistently smaller amplitude. Strikingly, only in the on-nerve recordings did these singing-related fluctuations survive common mode subtraction (Figure 5b, bottom; see also Figure S6a-b). This suggests that though there may be non-neuronal sources contributing to the nanoclip-recorded signals in this preparation, their effect is correlated across channels making them cleanly subtractable via virtual referencing. To quantify the relative strength of the neuronal and non-neuronal components of the recordings, we calculated the trial-by-trial SNR of singing-related signals from each device (Figure 5c, S6c), using singing-free epochs at the end of each recording to estimate the in vivo noise floor (Figure 5b, yellow band). Across animals, we found large differences in the SNR of on-nerve and off-nerve signals (Figure 5e, S6d). To quantify the temporal relationship between the recordings, we calculated the trial-by-trial correlation between the singing-related on-nerve and off-nerve signals (Figure 5d, S6d). Across animals, we found these correlations to be not significantly different from zero (Figure 5e). Taken together, these experiments suggest that the extent of artifact invasion in our chronic recording preparation is limited and are consistent with the 'on-nerve' nanoclip-recorded signals principally reflecting neuronal activity. On the small scale of the nanoclip, the six recording contacts show highly correlated movement and EMG artifacts and can thus be cleanly subtracted. In contrast, over the same length scale neural signals are much less correlated across electrodes and survive the common mode subtraction.

Minor criticisms

1. Editing for wordiness and run-on sentences is needed throughout the document.

We have revised the manuscript throughout to streamline language and reduce sentence length.

2. Other thin film nerve cuffs should be cited and discussed, such as TEENI (Desai et al 2017), tf-LIFE, TIME, and SELINE (Wurth et al.)

We thank the reviewer for highlighting these important contributions. We have revised the Introduction to include expanded discussion of these and other interfacing technologies. In addition, we now include the following references:

TEENI

Desai, V.H., Spearman, B.S., Shafor, C.S., Natt, S., Teem, B., Graham, J.B., Atkinson, E.W., Wachs, R.A., Nunamaker, E.A., Otto, K.J., et al. (2017). Design, fabrication, and characterization of a scalable tissue-engineered-electronic-nerve-interface (TEENI) device. In 2017 8th International IEEE/EMBS Conference on Neural Engineering (NER), (Shanghai, China: IEEE), pp. 203–206.

LIFE/tf-LIFE

*Lefurge, T., Goodall, E., Horch, K., Stensaas, L., and Schoenberg, A. (1991). Chronically implanted intrafascicular recording electrodes. *Ann Biomed Eng* 19, 197–207.*

*Bossi, S., Mencias, A., Koch, K.P., Hoffmann, K.-P., Yoshida, K., Dario, P., and Micera, S. (2007). Shape Memory Alloy Microactuation of tf-LIFEs: Preliminary Results. *IEEE Transactions on Biomedical Engineering* 54, 1115–1120.*

*Yoshida, K., Pellinen, D., Pivin, D., Rousche, P., and Kipke, D. (2000). Development of the thin-film longitudinal intra-fascicular electrode. In *Proceedings of the 5th Annual Conference of IFESS*, pp. p279-284.*

TIME

*Boretius, T., Badia, J., Pascual-Font, A., Schuettler, M., Navarro, X., Yoshida, K., and Stieglitz, T. (2010). A transverse intrafascicular multichannel electrode (TIME) to interface with the peripheral nerve. *Biosensors and Bioelectronics* 26, 62–69.*

SELINe

*Cutrone, A., Valle, J.D., Santos, D., Badia, J., Filippeschi, C., Micera, S., Navarro, X., and Bossi, S. (2015). A three-dimensional self-opening intraneural peripheral interface (SELINe). *J. Neural Eng.* 12, 016016.*

*Wurth, S., Capogrosso, M., Raspopovic, S., Gandar, J., Federici, G., Kinany, N., Cutrone, A., Piersigilli, A., Pavlova, N., Guiet, R., et al. (2017). Long-term usability and bio-integration of polyimide-based intra-neural stimulating electrodes. *Biomaterials* 122, 114–129.*

3. P-values are needed with all statements of significance in the results section.

We thank the reviewer for the recommendation. In this revision, we have included P-values with all statements of significance in the Results section. We note that details of the statistical analyses remain in the Methods section.

4. What was the stimulation threshold current used to produce the minimum evoked response in Figure 3?

Across $n = 3$ animals in the experiment, the stimulation threshold was $14.2 \pm 5.4 \mu\text{A}$. We have revised the manuscript to note this measure.

5. Correlation and mean amplitude plots in Figure 4D are unclear in the text and caption.

We have revised the figure, caption, and main text for clarity. At the recommendation of reviewers we have revised the analysis reported in 4D, eliminating the mean amplitude analysis entirely.

6. Can you describe the forces exerted on the nerve when pressed against the door via comparison to a similar technique? Similarly, fatigue characterization of the trap doors would be of interest. It should also be discussed whether the device can be removed from the nerve and repositioned during surgery.

We appreciate the opportunity to more fully describe the nanoclip, and we agree that additional detail about the mechanics and forces of implantation would be of interest to readers. To estimate the forces exerted on the nerve during implant, we modeled the trap door and hinge assembly based on the mechanical properties of the photoresist from which they are printed (i.e., IP-Dip). These simulations, reported in the new Figure S4, suggest that the force necessary to fully open the trap doors (which is borne by the nerve itself) is in the range of $1.25\text{-}7.5 \mu\text{N}$ (Figure S4b-d). Anchoring via elastically-deformable trap-doors is (to our knowledge) a novel concept in peripheral nerve interfacing, making it challenging to provide a direct comparison of these implantation forces with those of other interfaces. Nevertheless, several studies indicate that 30 mmHg ($\sim 4 \text{ kPa}$) is an important upper limit for nerve compression forces that can be tolerated chronically before onset of damage to nerve tissues and compromise of signaling function (Liu et al., 2018; Mackinnon, 2002; Rydevik et al., 1981). Given the size of the nanoclip and the small surface area in contact with the nerve during implant ($\sim 0.03 \text{ mm}^2$), this corresponds to an upper force limit of $\sim 120 \mu\text{N}$ – more than 1.5-2 orders larger than our simulations estimate are required to completely open the trap doors. We believe these mechanical simulations – in conjunction with the nerve functional assessment (Figure 2c,d) and the stability of chronic recordings (Figure 4) – support our claims that the implantation procedure and the associated forces are safe for the nerve and host tissues.

In addition, we used these mechanical models to analyze both deformation-related stresses in the hinges and to estimate the number of door-opening cycles to failure. We found that the forces necessary to fully open the doors ($1.25\text{-}7.5 \mu\text{N}$) produced maximum von Mises stresses of $< 0.25 \text{ MPa}$, well below the yield strength of the material (67 MPa ; Figure S4f).

Furthermore, these same simulations estimated lifetime of the trap doors at >15 full-loading/opening cycles before fatigue failure (Figure S4g). This implies – consistent with our experience using the device – that the elastically deformable doors are robust to surgical implantation forces.

Finally, we thank the reviewer for raising the issues of repositioning and reuse. We designed the nanoclip to prioritize ease of implant, precision fit to nerve, and long-term stability. A trade-off for achieving those aims was that we sacrificed some ability to reposition or reuse the device. Should a user wish to remove the nanoclip, there are two options: (1) use forceps to snap/shatter the printed anchor from the electrode (i.e., destroy the interface); (2) transect the nerve and pull the rump section through the device. Both removal options preserve the integrity of the electrodes for post-experiment inspection and characterization, but only the latter allows for the device to be re-implanted. Our experience working with the nanoclip and other peripheral interfaces suggests this is a worthwhile trade-off. We have revised the Results section to include these details.

7. It is unclear if the nanoclip can move along the length of the nerve. Is there an anchoring method used to stabilize the clip in a certain place?

We agree this is an important point that we did not discuss in our original submission. To reduce the potential for animal discomfort and incidental damage due to body dynamics, we used a single microsuture (placed approximately 1.5 cm from the electrode pads and nerve) to stabilize the ~35 mm interconnect in the body cavity. No other adhesive, sealant, suture, or other intervention was used to immobilize the nanoclip on the nerve. Nevertheless, the device is sized to be snug on the nerve and the authors have not observed any spontaneous movement or sliding of the nanoclip relative to the nerve during surgery. We have revised the Results to include this information.

8. Five animals were recorded until day 30, but 3 animals were recorded past the first month. For completeness, what was the reason for 2 animals not being recorded past the first month?

We thank the reviewer for pointing this out as it was indeed ambiguous. In the original manuscript, we reported chronically recording from $n = 5$ animals. Of those 5 animals, total recording durations were 17, 19, 31, 34, and 37 days. The 3 animals that were recorded for 30+ days were included in the analysis summarized in Figure 4. The two experiments that ran for less than 20 days ended with fracture/fatigue of the polyimide interconnects at approximately half the distance between the PCB (externalized at the head) and the nanoclip. We have revised the main text and methods sections to clarify both the durations of the chronic recording experiments and the early failure mode in these two instances.

9. It is important to see the examples of fictive vocalizations in Figure 5e, but the figure is extremely small and hard to see.

We thank the reviewer for this suggestion. We have reorganized Figure 5 (in this submission, now Figure 7) in part to enlarge the fictive vocalization examples.

10. Is the reason behind the lower stimulation threshold for the nanoclip than the bipolar silver hook the better insulation provided by the nanoclip?

It is challenging to investigate this issue directly, but that is our supposition. By creating a clip that is snug around the nerve, we hypothesize that we have increased the electrical isolation of the electrode pads from the surrounding extra-neural environment. In this scenario, there would be reduced current leakage at the nanoclip-nerve interface, allowing a greater fraction of the injected charge to depolarize the tissue rather than dissipate into the extra-neural environment. Thus, we believe the lower stimulation thresholds we report are a reflection of increased efficacy of charge transfer to the nerve rather than any novel mode of stimulation or physiological change in excitability of the nerve. We have revised the text to include this point.

Reviewer #2

This paper presents a novel microscale implantable device capable of recording from and stimulating small nerves. The engineering feats are coupled with detailed electrophysiological and behavioral analysis in the songbird vocal system. While a simpler version of this devices has been published earlier (Lissandrello et al., 2017), this device is novel as it incorporates a microfabricated, multi-channel thin film electrode substrate, which allows for more precise stimulation and recording than the previous version. In addition, this paper presents chronic behavioral and electrophysiological data, which has not before been published.

In general, the data presented in this paper constitute a ground-breaking advance in small nerve recording and modulation, and provides a tool for more detailed study of nerve signals and function than currently exists. However, figures tend to lack detail, particularly in the realm of data analysis, and should be improved prior to publication. A list of major concerns is presented below:

Major criticisms

1. Figure 2 – Histology is mentioned in the text, but no histological images or quantitative assessment is provided. Please provide evidence of nerve integrity with implantation of this device, both qualitative and quantitative.

We thank the reviewer for raising this issue. As we noted in response to Reviewer 1 above, we agree that additional data and quantitative analyses are required to support any histological claims of reduced inflammatory response and fibrotic penetration post-implantation. Prompted in part by these requests, we are planning experiments to address the long term bio-integration and safety of the nanoclip in a clinically relevant model (i.e., rats) with established protocols for quantifying implantation-related inflammation, fibrosis, and nerve integrity. Though important, these additional experiments and analyses are beyond the scope of the current manuscript and will be reserved for a future report. In this

revised manuscript, we have removed references to histological results and simply note that “In all chronically implanted animals for which necropsies were performed (n = 5 birds), the nanoclips were found securely latched over the nerve up to 11 months after implant, suggesting the potential for long-term viability of the preparation.”

Nevertheless, we believe the experiments we report speak directly to the issue of nerve integrity following implant. In the nerve function studies reported in Figures 2c-d and S5, we demonstrate that bilateral nXIIIs nanoclip implants produce no significant changes in the acoustic structure of song for up to 8 days following the procedure. Given that song acoustic structure requires precise control of the syrinx (Goller and Riede, 2013; Goller and Suthers, 1996) and that even an acute crush injury can generate significant and prolonged changes in song structure, the absence of song change following implant suggests that the procedure and chronic presence of the nanoclip does not impair nerve function. Though we appreciate that this analysis cannot substitute for a histological assessment of biointegration and host tissue health, we believe these results support our conclusion that the nanoclip can be implanted without significant disruption of nerve function. This is now further elaborated in the text.

2. Figure 2B- Please provide quantitative analysis of the different in spectral similarity between nerve crush, sham and Nanoclip implant from non-injured baseline similarity.

We appreciate the opportunity to further clarify our analysis of the nerve function assessment data in Figure 2. As requested, we include quantification of the change in acoustic similarity from baseline to days 1 and 8 post-implant for each treatment group. Neither the nanoclip-implanted, sham implant, nor intact control experimental groups produced songs that were significantly acoustically different from uninjured baseline at early or late timepoints; birds receiving bilateral nerve crush injuries were significantly different on both days. We have revised the text and added Figure 2c to report this new analysis. To further illustrate the effect of these manipulations on song structure, we now add Figure S5 showing representative spectrograms of song motifs produced before and 1 day after each manipulation.

3. Figure 4 presents evidence for the stability of nerve recordings across 30 days. However, the metric used to quantify these recordings, a smoothed activity envelope, is atypical for neural recordings. While not improper, a more conventional metric would be more meaningful for future neurophysiological applications of this technology. For instance, Zanos et al. recently presented vagal nerve recordings that used wavelet decomposition to identify unique compound action potential signals (CAPS), and then reported event rate, inter-cap-interval histograms and signal-to-noise ratio for each CAP, metrics that are easily comparable to single unit recordings from the brain (Zanos et al., 2018). Given the author’s background in neurophysiological recordings in the songbird (ie. Ölveczky et al., 2011)), it is surprising that a more detailed analysis of neurophysiological data was not applied. I would recommend the authors add additional neurophysiological analysis to demonstrate that this technology can be meaningful to the songbird field, and more broadly.

We completely agree with the reviewer on the import of interpretable metrics for comparison to other technologies. In this resubmission, we have revised the analysis of the recording data to provide additional measures of stability over time. Specifically, we have adapted this reviewer's suggestion (from comment #4 below) and have added more conventional analyses of the signal-to-noise ratio and event rate for singing-related activity – shown both trial-by-trial in a representative animal (Figure 4d) and mean-per-day for n = 3 animals (Figure 4e). Across animals, neither metric shows significant change over 30 days, further supporting our assessment of chronic recording stability.

In addition, we have retained in this revision the smoothed activity envelope as but one of three metrics demonstrating chronic stability. Though we acknowledge that this is not the most commonly used metric, we do believe there are experimental contexts, recording types, and analyses to which it is well-suited to measure and track the temporal structure of neuronal activity. In particular, multi-unit recordings from independent source-dense nervous system structures with high firing rates often exhibit significant superposition of action potential waveforms, making analyses reliant on identifying individual spikes less robust. Thus in prior multi-unit recordings from the songbird motor system, where singing-related firing rates often exceed 200 Hz, we and others have shown that smoothed activity envelopes are robust, informative readouts of neuronal population dynamics and underlying function (Ali et al., 2013; Day et al., 2008; Otchy et al., 2015). In addition, similar activity envelopes are frequently reported for chronic recordings from the peripheral sympathetic nervous system in rodents (Stocker and Muntzel, 2013). Thus, we believe there is merit and utility in retaining the activity envelope correlations as one measure of chronic stability.

Finally, we agree that decomposition methods are powerful tools for analyzing recording data, and we are admirers of the analysis from ref. (Zanos et al., 2018) the reviewer cites. Indeed as the reviewer mentions, the author has experience with these decomposition methods to analyze single-unit singing-related activity in the songbird motor system (Ölveczky et al., 2011). The Zanos analysis – like many decomposition/spike-sorting methods (Aljadeff et al., 2016; Jackel et al., 2011; Quiroga et al., 2004) – begins with action potential detection via threshold crossing. For recordings from nervous system structures with low average firing rates and/or sparse high-SNR events (like those in Zanos et al), thresholding is a reliable detection method. However, in dense multi-unit recordings where independent, high firing rate events non-uniformly overlap – like recordings from the songbird motor system – these techniques have in our hands been less effective in detecting event waveforms that can be meaningfully clustered in later stages of analysis. Though we recognize the analytical insight that may be gained by successfully applying decomposition techniques to our recordings, differences in the structure of the neuronal activity they contain make this non-trivial and a significant research project in itself that is beyond the scope of this manuscript. Nevertheless, we believe that the data and metrics we present in this revised manuscript are sufficient to support our statements of chronic recording stability.

4. The longevity data in Figure 4 looked very promising and exciting. Again, I would have preferred to see this data quantified more extensively – ie. Signal-to-noise, event rate. Predictive value for syllable shape would be amazing, although not required.

We are grateful for the suggestion of how to reanalyze our chronic recording data to strengthen our manuscript. As recommended, we now report in Figure 4 trial-by-trial signal-to-noise ratios and event rates for singing related nerve activity. We show this both in a representative animal (Figure 4d) and across experiments (Figure 4e; n = 3). Importantly, this reanalysis did not impact any of our conclusions or statements of significance concerning the chronic recording stability of the nanoclip.

5. Figure 5 shows specificity of stimulation across different contact sites. The variability is quantified by tSNE plot. While tSNE offers the advantages of being an unsupervised non-linear visualization technique, it can be prone to misinterpretation, and can change shape with differing perplexity and step values. In particular, distances between and within clusters can change dramatically with different iterations. Replication of the results in Fig. 5F,G using more conventional syllable variability metrics are necessary to support tSNE measurements, and the overall conclusion regarding stimulation specificity.

We completely agree with the reviewer that distances in a low-dimensional tSNE-space are prone to misinterpretation and can be highly sensitive the choice of embedding parameters. We report the complete set of t-SNE embedding parameters (perplexity, pre-processing dimensional reductions, distance metrics, etc.) in the Methods to improve the interpretability and replicability of the visualization in Figure 5 (in this revision, Figure 7). Nevertheless, we agree that quantitative claims of stimulation specificity would be strengthened with a more conventional syllable variability metric.

In this revised submission, we replaced the mean pairwise t-SNE distance analysis with the acoustic similarity metric (Tchernichovski et al., 2000) used in the post-implantation functional analysis (Figure 2c-d) as well as in prior studies of song learning and variability (Balmer et al., 2009; Kozhevnikov and Fee, 2007; Ölveczky et al., 2011; Ravbar et al., 2012). We compared the mean acoustic similarity of vocalizations elicited by a given stimulation pattern with those produced by the other stimulation patterns. Across n=6 animals, we found that fictive vocalizations produced by the same stimulation pattern are significantly more similar to each other than they are to those produced by other stimulation patterns. For comparison, we performed the same acoustic similarity analysis on natural vocalizations from an additional n= 6 birds, using semi-automated annotation of spectrograms to define ‘same’ and ‘different’ syllable types. (Please see Methods for details on the annotation procedure.) As expected, this analysis showed that syllables are more similar to those of the same type than they are to those of a different type. Strikingly, this analysis also demonstrated that naturally produced syllables of different types were more similar to each other than were fictive syllables produced by different stimulation patterns. This suggests not only that the nanoclip is capable of precise functional stimulation of a small peripheral nerve, but also that it is capable of producing nerve activity states that are at least as dissimilar from each other than those occurring during natural behavior.

Minor criticisms

Fig. 4D – please provide a color scale for the color map.

We thank the reviewer pointing this out – we have added a color scale bar in the revised manuscript.

Fig. 4D – the text box overlaps the figure labels.

We have corrected this formatting error in this resubmission.

Fig. 4F – does this refer to the signal envelope, or the raw (processed?) signal?

We regret the ambiguity – this figure reports the signal envelope. We have revised the caption and text to clarify.

Reviewer #3

This exciting manuscript presents a new device/system for recording and manipulating the activity of small peripheral nerves, in vivo, during behavior and over periods as long as 1 month. Small-scale engineering and experimental neurobiology are brought together to develop small, flexible, electrode arrays that are secured to the surface of an intact nerve such that compound action potentials can be recorded and nerve fibers can be stimulated in small, freely behaving animals during behavior. The function of this new interface is demonstrated with multiple experiments. Because development of such systems are important for understanding how neural signals in the brain are transformed by the peripheral nervous system into muscle activity, the results of this study will be impactful for experimental and clinical neuroscience. The manuscript is well written; the system components, experiments, data and analyses are clearly described. Below, I suggest improvements to the manuscript that would further demonstrate the system's safety and broad-scale usefulness.

Major criticisms

The choice of the zebra finch for testing the system is wise because the anatomy and physiology of the PNS involved in singing are well described and the motor production of song is highly stereotyped. Here, the tracheosyringeal nerve, which innervates the muscles of the syrinx, and singing behavior are studied to test the usefulness of the system. First, the authors show that they reliably obtained interpretable nerve recordings over time (indicating little to no nerve damage) and across animals (indicating precise and consistent placement of electrode contacts). Second they show that the system can be used to evoke nerve activity via electrical stimulation that is specific to the spatial pattern of stimulating electrode contacts, and that results in vocal output.

1. While these demonstrations are strong, they could be improved without collecting more data. First, the results of the histological analyses of nerve segments that were housed in the clip for weeks should be added. If the quantified analyses of control and clipped nerve segments shows that the clip does not damage the nerve fibers even after a month of implantation, then the success of the clip design will be strongly supported.

We agree with the reviewer that additional quantitative histological analyses of nerve segments in and out of the nanoclip would strengthen support for our design and strengthen our manuscript. In the original submission, we provided

a qualitative description of the histology response as low sectioning yield (due to frequent fracture of the acrylic nanoclip) limited thorough quantitative analysis. In addition, the songbird model used here lacks well-defined immunohistological markers for peripheral inflammation and fibrotic tissue growth, making it poorly suited to histological assessments of nerve fiber health. As we recognize the importance of these experiments to gaining a complete picture of device-tissue interactions, we are planning experiments to address the long term bio-integration and safety of the nanoclip in a model suitable for these investigations (i.e., rats). Though important, these additional experiments and analyses are beyond the scope of the current manuscript and will be reserved for a future report. In this revised manuscript, we have removed references to histological results and simply note that “In all chronically implanted animals for which necropsies were performed (n = 5 birds), the nanoclips were found securely latched over the nerve up to 11 months after implant, suggesting the potential for long-term viability of the preparation.”

Nevertheless, we believe the experiments we report speak directly to the issue of nerve integrity following implant. In the nerve function studies reported in Figures 2c-d and S5, we demonstrate that bilateral nXIIIs nanoclip implants produce no significant changes in the acoustic structure of song for up to 8 days following the procedure. Given that song acoustic structure requires precise control of the syrinx (Goller and Riede, 2013; Goller and Suthers, 1996) and that even an acute crush injury can generate significant and prolonged changes in song structure, the absence of song change following implant suggests that the procedure and chronic presence of the nanoclip does not impair nerve function. Though we appreciate that this analysis cannot substitute for a histological assessment of biointegration and host tissue health, we believe these results support our conclusion that the nanoclip can be implanted without significant compromise of nerve function. This is now further clarified in the text.

2. Second, the Results should include information on the estimated number and spatial extent of fibers that are stimulated by each electrode contact and how that relates to the experimental observations of different activity patterns and vocalizations when different spatial patterns of stimulation are used. The addition of this information will strengthen the argument that the system can be used to manipulate nerve activity in a precise manner.

This is an interesting point that we did not address in the initial submission, and we are grateful for the reviewer’s suggestion that we do. Our prior studies reveal that the nXIIIs is composed of ~1000 nerve fibers, each 4-6 μm in diameter (Gillis et al., 2018; Lissandrello et al., 2017). If we assume uniform distribution of fibers and straight fiber paths within the implanted segment of nerve, the number of fibers within a given distance of each electrode pad can be calculated directly – here, ~80 fibers (8% of total) within are 20 μm and ~335 (33% of total) are within 50 μm .

Estimating the spatial extent of fiber recruitment during multi-channel stimulation is considerably more difficult as it is a function of current paths through the tissue that are highly dependent on the interaction of the stimulation parameters (current direction, amplitude, pulse width, and relative position of active electrodes) with local inhomogeneities in the nerve tissue (Hokanson et al., 2018; Pelot et al., 2018). Furthermore, while there are several detailed studies of fiber recruitment in large diameter nerves and devices with millimeter-plus electrode spacing, we are not aware of any similar

studies of small nerves and interfaces with 10s of microns between microelectrodes. Nevertheless, inferring from prior studies (Grill and Mortimer, 1996; Veltink et al., 1988), we conservatively estimate that fibers within 50 μm of the electrode are likely to be affected in these experiments. We have revised the text to reflect these estimates.

3. The broad scale usefulness of a new approach/system is clearest when the system is tested in more than one experimental context, here, a nerve. With the goals in the Introduction in mind, the paper would be stronger if clipping, recording and stimulating were demonstrated in a second nerve. A candidate is the 8th nerve, which is accessible and reasonably well studied in birds. The addition of these data would show the generalizability of implant placement without functional damage, accurate recording of nerve activity and the effectiveness of stimulation.

We very much appreciate the spirit of this comment as we agree that testing in additional experimental contexts is an important way to demonstrate broad applicability. Indeed, it was a desire to validate in multiple contexts that prompted our move beyond the canonical acute recording and stimulating assays and to instead establish utility in more technically challenging chronic recording and 'fictive singing' preparations. Taken together, we believe these experiments multiply demonstrate the nanoclip's recording and stimulating capabilities and functional safety.

Though the reviewer is correct that replicating these experiments in other nerves could strengthen the manuscript – more data and evidence are rarely a negative – we do not believe that new experiments in additional targets are necessary at this stage to support the claims in the manuscript. Furthermore, given COVID-related research suspensions and several project staff moving to new positions, we do not have the resources to develop new preparations replicating these studies in other nerves. Nevertheless, as we believe the nanoclip fills an important niche in interfacing technology for small nerves, we are now pursuing additional funding to support such studies in new nerves and model systems, and we anticipate devoting much of our future effort to expanding access to this promising new interfacing technology.

Minor criticisms

1. Add detection threshold to Figure 4c.

We thank the reviewer for the opportunity to clarify. The automated recording software we used was triggered by a detection threshold applied to the audio signals from the in-cage microphone. Specifically, thresholds were set to detect periods in which power in the 2500 - 8000 Hz band (corresponding to song) exceeded 10-50 times the power in the 50 - 250 Hz band (corresponding to low frequency background noise). Thus, there were no detection thresholds applied to the nXIIIts recordings appearing in Figure 4c. We regret ambiguity in our description, and we have revised the Methods to clarify how recordings were acquired.

2. Add one high resolution plot of the smoothed activity for at least 1 ABC sequence to Figure 4d. The data shown as a heatmap over recording days is great, but an intermediate step between panels c and d is needed.

We thank the reviewer for the suggestion to show the smoothed activity for the ABC traces. We have revised Figure 4c to show the smoothed activity envelopes (in gray) for each of the representative recordings shown on Days 1, 10, 20, and 30.

3. If I understand the Figure 5 experiments, panels a – c and d – g describe completely different experiments. The top panels show experiments with 2 clips, one recording and one stimulating. The bottom panels show experiments using only a stimulating clip, and vocal production with stimulation of the nerve and air pressure in the respiratory system. If this is correct, then Figure 5 should be revised into 2 figures, each showing one of these experiments.

The reviewer is correct – these panels describe different acute stimulation experiments. As suggested, we have revised these panels in to 2 separate figures (Figure 6 and Figure 7).

4. The full paragraph on the second page of the Discussion includes sentences, phrases and words that systems neuroscience readers with not understand without explanation. Examples are “present elastic moduli substantially above the kilopascal range of the host tissue” and “despite the Young’s modulus of their bulk material being within the gigapascal range.” Please add explanation.

We thank the reviewer for identifying these areas of confusion and complexity. We have revised this section with an eye toward readability for a scientifically-informed but not necessarily engineering-oriented audience.

5. The Methods section includes a subsection on the quantification of 5 syllable features. But these features are not in the Results.

We thank the reviewer for pointing this out as it was indeed unclear how this analysis was related to the data presented. The song similarity analysis reported in Figure 2, and described originally in ref. (Tchernichovski et al., 2000), begins with quantifying the median Euclidean distance between two song syllables in the 5-dimensional space defined by these acoustic features. This distance is then converted to a similarity P-value (ranging from 0-1, with 1 indicating identity), which is the metric ultimately reported in Figure 2c-d. In addition, in this revised manuscript we use the same analysis to quantify the similarity of both natural zebra finch syllables and fictive vocalizations produced by the same or different multi-channel stimulation patterns (Figure 7d). We regret the ambiguity and have revised the Methods section to clarify the role of acoustic feature quantification.

6. Typo in the Methods “was calculated was calculated”

We thank the reviewer for identifying this typo. We have corrected this in the revision.

7. Typo in the Methods “20 unrelated.”

We thank the reviewer for identifying this typo. We have corrected this in the revision.

8. Add page numbers or line numbers to the ms

As suggested, we have added page numbers to the manuscript.

References

- Ali, F., Otchy, T.M., Pehlevan, C., Fantana, A.L., Burak, Y., and Ölveczky, B.P. (2013). The Basal Ganglia Is Necessary for Learning Spectral, but Not Temporal, Features of Birdsong. *Neuron* *80*, 1–13.
- Aljadeff, J., Lansdell, B.J., Fairhall, A.L., and Kleinfeld, D. (2016). Analysis of Neuronal Spike Trains, Deconstructed. *Neuron* *91*, 221–259.
- Balmer, T.S., Carels, V.M., Frisch, J.L., and Nick, T.A. (2009). Modulation of Perineuronal Nets and Parvalbumin with Developmental Song Learning. *J. Neurosci.* *29*, 12878–12885.
- Day, N.F., Kinnischtzke, A.K., Adam, M., and Nick, T.A. (2008). Top-Down Regulation of Plasticity in the Birdsong System: “Premotor” Activity in the Nucleus HVC Predicts Song Variability Better Than It Predicts Song Features. *J. Neurophysiol.* *100*, 2956–2965.
- Gillis, W.F., Lissandrello, C.A., Shen, J., Pearre, B.W., Mertiri, A., Deku, F., Cogan, S., Holinski, B.J., Chew, D.J., White, A.E., et al. (2018). Carbon fiber on polyimide ultra-microelectrodes. *J. Neural Eng.* *15*, 016010.
- Goller, F., and Riede, T. (2013). Integrative physiology of fundamental frequency control in birds. *J. Physiol.-Paris* *107*, 230–242.
- Goller, F., and Suthers, R.A. (1996). Role of syringeal muscles in gating airflow and sound production in singing brown thrashers. *J. Neurophysiol.* *75*, 867.
- Grill, W.M., and Mortimer, J.T. (1996). Quantification of Recruitment Properties of Multiple Contact Cuff Electrodes. *IEEE Trans. Rehabil. Eng.* *4*, 49–63.
- Hokanson, J.A., Gaunt, R.A., and Weber, D.J. (2018). Effects of Synchronous Electrode Pulses on Neural Recruitment During Multichannel Microstimulation. *Sci. Rep.* *8*, 13067.
- Jackel, D., Frey, U., Fiscella, M., and Hierlemann, A. (2011). Blind source separation for spike sorting of high density microelectrode array recordings. (*IEEE*), pp. 5–8.
- Kozhevnikov, A.A., and Fee, M.S. (2007). Singing-Related Activity of Identified HVC Neurons in the Zebra Finch. *J. Neurophysiol.* *97*, 4271–4283.
- Lissandrello, C.A., Gillis, W.F., Shen, J., Pearre, B.W., Vitale, F., Pasquali, M., Holinski, B.J., Chew, D.J., White, A.E., and Gardner, T.J. (2017). A micro-scale printable nanoclip for electrical stimulation and recording in small nerves. *J. Neural Eng.* *14*, 036006.
- Liu, Z.-Y., Chen, Z.-B., and Chen, J.-H. (2018). A novel chronic nerve compression model in the rat. *Neural Regen. Res.* *13*, 1477.

- Mackinnon, S.E. (2002). Pathophysiology of nerve compression. *Hand Clin.* 18, 231–241.
- Ölveczky, B.P., Otchy, T.M., Goldberg, J.H., Aronov, D., and Fee, M.S. (2011). Changes in the neural control of a complex motor sequence during learning. *J. Neurophysiol.* 106, 386–397.
- Otchy, T.M., Wolff, S.B.E., Rhee, J.Y., Pehlevan, C., Kawai, R., Kempf, A., Gobes, S.M.H., and Ölveczky, B.P. (2015). Acute off-target effects of neural circuit manipulations. *Nature* 528, 358–363.
- Pelot, N.A., Thio, B.J., and Grill, W.M. (2018). Modeling Current Sources for Neural Stimulation in COMSOL. *Front. Comput. Neurosci.* 12.
- Quiroga, R.Q., Nadasdy, Z., and Ben-Shaul, Y. (2004). Unsupervised spike detection and sorting with wavelets and superparamagnetic clustering. *Neural Comput.* 16, 1661–1687.
- Ravbar, P., Lipkind, D., Parra, L.C., and Tchernichovski, O. (2012). Vocal Exploration Is Locally Regulated during Song Learning. *J. Neurosci.* 32, 3422–3432.
- Rydevik, B., Lundborg, G., and Bagge, U. (1981). Effects of graded compression on intraneural blood flow. *J. Hand Surg.* 6, 3–12.
- Stocker, S.D., and Muntzel, M.S. (2013). Recording sympathetic nerve activity chronically in rats: surgery techniques, assessment of nerve activity, and quantification. *Am. J. Physiol.-Heart Circ. Physiol.* 305, H1407–H1416.
- Tchernichovski, O., Nottebohm, F., Ho, C.E., Pesaran, B., and Mitra, P.P. (2000). A procedure for an automated measurement of song similarity. *Anim. Behav.* 59, 1167–1176.
- Veltink, P.H., van Alste, J.A., and Boom, H.B.K. (1988). Influences of stimulation conditions on recruitment of myelinated nerve fibres: a model study. *IEEE Trans. Biomed. Eng.* 35, 917–924.
- Zanos, T.P., Silverman, H.A., Levy, T., Tsaava, T., Battinelli, E., Lorraine, P.W., Ashe, J.M., Chavan, S.S., Tracey, K.J., and Bouton, C.E. (2018). Identification of cytokine-specific sensory neural signals by decoding murine vagus nerve activity. *Proc. Natl. Acad. Sci.* 115, E4843–E4852.

REVIEWERS' COMMENTS:

Reviewer #1 (Remarks to the Author):

The paper is much improved, and the majority of my comments have been addressed. There are still two issues that should be strongly considered prior to publication.

First, the addition of new control experiments in Figure 5 and S6, showing that the signal can be recorded from on the nerve but not immediately off the nerve is compelling. I am also very encouraged that the authors are very concerned about the possibility of artifact. My concern that this is artifactual still remains, and the comparison table highlights why. The closest similar design cited is Gonzalez-Gonzalez et al, where the present study states in error that they report 400 uVpp spontaneous signals. They actually report closer to 100 uV signals, which is particularly good for a nerve cuff. While the form factor, implantation method, etc are very different from other nerve cuffs, from an electrical perspective one might expect them to be very similar. It also appears that my question about the average spontaneous chronic Vpp across n=? birds was not answered in the response, and the 500 uV referred to throughout the paper could be exemplar, which is not appropriate.

At minimum here, there needs to be a line of mechanistic, physics based reasoning about why these signals are so much larger than other devices. These signals would need to be larger still in the extracellular space within the epineurium. Is that likely to be true for some reason? The one sentence listing various differences isn't enough. You should also consider the possibility of EMG invasion from the innervated muscle into the fluid of the nerve. I strongly recommend language in the Discussion suggesting that this may not ultimately be ENG., i.e. axonal recordings To summarize, the control experiments are exactly what one would think to do, but this paper could still become famously artifactual, and you may wish to hedge your interpretation of the results.

Second, all reviewers had real concerns about the histological results, and I am still very concerned that exemplar data is being reported without being labeled as such. There is no n given in the Methods paragraph on immunohistology. If immunohistology is outside the scope of this paper, which is a reasonable argument given the length, it should just be removed. If not, you should make a good faith effort to characterize other devices, report an n of attempts, and whether or not contradictory data to Figure 2b was ever observed. Cracked sections may not be good for figures, but they can still be evaluated in many cases.

Reviewer #2 (Remarks to the Author):

The authors have meaningfully and compellingly addressed my concerns, and added in a figure of significant value to address possible EMG contamination. These additions greatly strengthen the paper, and demonstrate the novelty and utility of this approach. Now, if only these authors can start widely distributing this device, so that I can use it in my research! I recommend no further revisions to this manuscript.

Cristin Welle

Reviewer #3 (Remarks to the Author):

As previously described, the technical developments presented in this manuscript are potentially impactful. The Reviewers put significant effort into evaluating the potential value of the nanoclip device and clearly communicated the additions that would convincingly support the use and further development of the nanoclip. Examples were showing the device's feasibility as a chronic implant and its potential for application beyond one nerve, in one system, in one animal. Unfortunately, the authors decided not to address the Reviewers' major comments and therefore didn't significantly improve the manuscript.

Response to reviewer and editorial comments

We thank the reviewers and editors for close reading and analysis of our revised manuscript. We are gratified that the reviewers generally found our additions responsive to their concerns and the manuscript improved from our initial submission. In addition, we are grateful for the opportunities to address further the residual concern of Reviewer 1 and to bring our manuscript into align with the editorial and formatting standards of Nature Communications.

Reviewer #1

The paper is much improved, and the majority of my comments have been addressed. There are still two issues that should be strongly considered prior to publication.

First, the addition of new control experiments in Figure 5 and S6, showing that the signal can be recorded from on the nerve but not immediately off the nerve is compelling. I am also very encouraged that the authors are very concerned about the possibility of artifact.

We are gratified that the reviewer found our additional control studies compelling, and we thank the reviewers and editors for the opportunity to include these important results.

My concern that this is artifactual still remains, and the comparison table highlights why. The closest similar design cited is Gonzalez-Gonzalez et al, where the present study states in error that they report 400 uVpp spontaneous signals. They actually report closer to 100 uV signals, which is particularly good for a nerve cuff.

We appreciate the reviewer raising this issue as we believe it highlights some ambiguity in how we reported the estimates of recording quality in Gonzalez-Gonzalez et al in Table 1. There are several experiments in this excellent paper, but for reasons of both concision and comparability to the preparations we report in this manuscript we listed two estimates of recording peak-to-peak voltage: one for acute recordings of pelvic nerve during bladder filling ($V_{pp} = 150 \mu V$), and one for subchronic recordings of sciatic nerve during electrical stimulation ($V_{pp} = 400 \mu V$). We did not report on spontaneous signals from this paper. For the former, our estimate was drawn from the authors' statement, "During the increase in vesicular pressure, we recorded high frequency CNAP activity of approximately 157 μV peak-to-peak over the noise starting at 3–5 mmHg in vesicular pressure." For the latter, the authors did not quantify peak-to-peak voltage explicitly in the text, thus our estimate was based on the examples of stimulation-evoked recordings in Figure 7D. Even with care, estimating values from lightly-labeled figures is challenging, but we believe that $V_{pp} = 400 \mu V$ is a reasonable estimate.

Nevertheless, we acknowledge that the data in Table 1 could have been presented more clearly. In this revision, we have reformatted Table 1 to improve readability and have expanded the legend to clarify the data reported.

While the form factor, implantation method, etc are very different from other nerve cuffs, from an electrical perspective one might expect them to be very similar. It also appears that my question about the average spontaneous chronic V_{pp} across $n=?$ birds was not answered in the response, and the 500 uV referred to throughout the paper could be exemplar, which is not appropriate.

We thank the reviewer for raising again the issue of chronic recording peak-to-peak voltages. In the interest of balancing competing reviewer requests, we removed the normalized peak-to-peak voltages from the original submission and replaced it with additional metrics that were suggested by another reviewer. As noted in our reply, we believed this would be responsive to this reviewer's concern, and we regret this was not the case.

In this revision, we include additional plots (Figure 4d,e) reporting the peak-to-peak voltages in the chronic recording experiments ($n = 3$ birds in total), and we have revised the text accordingly. We now state explicitly, mean peak-to-peak voltage across all days (Days 1, 10, 20, and 30) and birds ($n = 3$) $438 \pm 64 \mu V$ (range of 334 – 522 μV). In addition, mean daily peak-to-peak voltages exceeded 500 μV on at least one day in 2 of 3 birds, consistent with our estimate of "up to 500 μV ". Importantly, we note this additional analysis does not alter

our basic assessment of chronic recording performance. We hope this addresses the reviewer's remaining concerns.

At minimum here, there needs to be a line of mechanistic, physics-based reasoning about why these signals are so much larger than other devices. These signals would need to be larger still in the extracellular space within the epineurium. Is that likely to be true for some reason? The one sentence listing various differences isn't enough. You should also consider the possibility of EMG invasion from the innervated muscle into the fluid of the nerve. I strongly recommend language in the Discussion suggesting that this may not ultimately be ENG., i.e. axonal recordings. To summarize, the control experiments are exactly what one would think to do, but this paper could still become famously artifactual, and you may wish to hedge your interpretation of the results.

We thank the reviewer for the opportunity to elaborate further on the quality of the chronic recordings reported in this manuscript. At the outset, we note that signals of this scale have been described in acute and subchronic recordings of evoked responses, some of which are cited in Table 1 (Elyahoodayan et al., 2020; González-González et al., 2018; Lee et al., 2016; Seo et al., 2016). Thus, what is called for is a line of reasoning about maintaining this scale of signal in a chronic recording of spontaneous activity. Our thinking here principally follows two paths – either of which (or some combination of both) are plausible explanations.

First, the design, manufacturing tolerances, and fit on the nerve contribute to a physics-based hypothesis of signal recording quality – namely, that by closely fitting a device to the nerve we reduce a high-conductivity path (provided by saline/extra-neural fluids) between the electrode-nerve interface (within the device) and the external environment. Though we did not experimentally test this hypothesis in the present work, this explanation is consistent with prior modeling (Struijk, 1997) and in vivo (Andreasen et al., 2000) studies demonstrating that reducing conductivity through a nerve cuff can significantly increase peak-to-peak voltages. In addition, we speculate from prior studies that the small size of the device, highly flexible interconnect, and closeness of fit may play some role in slowing or reducing the local reactive tissue responses that over time envelop electrode pads within the device and degrade recorded signals (Grill and Mortimer, 2000; Onuki et al., 2008; Salatino et al., 2017; Ward, 2008). Each of these are potentially important factors to be considered in future studies.

Second, it is possible that nXIIIts neurophysiology and singing-related activity generate spontaneous voltage fluctuations within the nerve that are larger than are typical in more common implantation targets like the sciatic, splanchnic, or pelvic nerves. As the signal recorded with a cuff-like electrode is (in part) a function of the super position of nearby active current sources within the nerve (Andreasen et al., 1997), both the number of co-active nerve fibers and the frequency of action potentials traversing them can affect the size of recorded signals. Sciatic, pelvic, and splanchnic recruitment vary by task, but single fiber firing rates are typically <50 Hz (Musick et al., 2015; Su et al., 2007; Yu and de Groat, 2008). We are not aware of prior studies characterizing singing-related nXIIIts activity, but recordings from other avian brain motor regions suggest very high firing rates (>>200 Hz) and a large fraction of co-active units are common during singing (Fee et al., 2004; Leonardo and Fee, 2005; McCasland, 1987; Ölveczky et al., 2011). Thus, it is plausible that these physiological differences contribute to the large spontaneous signals we report. Nevertheless, the key observation is not that the nanoclip yields unusually large signals, but rather that these signals remain stable over the timescale of weeks. Speculation about high firing rates or synchrony does not provide an explanation for stable performance. Instead, we believe the best explanation is that few chronic recordings from small nerves have been reported previously, and that with devices precisely matched to their size, high-SNR signals may be more accessible than has been anticipated. If true, this is good news for the study of small nerve signaling in the periphery.

As suggested, we have revised the discussion section to include this more detailed account of our reasoning.

Second, all reviewers had real concerns about the histological results, and I am still very concerned that exemplar data is being reported without being labeled as such. There is no n given in the Methods paragraph on immunohistology. If immunohistology is outside the scope of this paper, which is a reasonable argument given the length, it should just be removed. If not, you should make a good faith effort to characterize other devices, report an n of attempts, and whether or not contradictory data to Figure 2b was ever observed. Cracked

sections may not be good for figures, but they can still be evaluated in many cases.

In this revision, we have removed all mention, description, and depiction of immunohistology.

Reviewer #2

The authors have meaningfully and compellingly addressed my concerns, and added in a figure of significant value to address possible EMG contamination. These additions greatly strengthen the paper, and demonstrate the novelty and utility of this approach. Now, if only these authors can start widely distributing this device, so that I can use it in my research! I recommend no further revisions to this manuscript.

Cristin Welle

We thank the reviewer for her time and effort reviewing our manuscript.

Reviewer #3

As previously described, the technical developments presented in this manuscript are potentially impactful. The Reviewers put significant effort into evaluating the potential value of the nanoclip device and clearly communicated the additions that would convincingly support the use and further development of the nanoclip. Examples were showing the device's feasibility as a chronic implant and its potential for application beyond one nerve, in one system, in one animal. Unfortunately, the authors decided not to address the Reviewers' major comments and therefore didn't significantly improve the manuscript.

We thank the reviewer for the time and effort spent reviewing our manuscript. Of the 11 comments and recommendations the reviewer provided, we fully adopted/addressed 9 of them. Though we appreciate the spirit and aim of the 2 unfulfilled requests, we maintain that the additional experiments are unnecessary to support our principal claims and are thus beyond the scope of this manuscript.

References

Andreasen, L.N.S., Struijk, J.J., and Haugland, M. (1997). An artificial nerve fiber for evaluation of nerve cuff electrodes. In *Proceedings of the 19th Annual International Conference of the IEEE Engineering in Medicine and Biology Society. "Magnificent Milestones and Emerging Opportunities in Medical Engineering"* (Cat. No.97CH36136), pp. 1997–1999 vol.5.

Andreasen, L.N.S., Struijk, J.J., and Lawrence, S. (2000). Measurement of the performance of nerve cuff electrodes for recording. *Med. Biol. Eng. Comput.* 38, 447–453.

Elyahoodayan, S., Larson, C., Cobo, A.M., Meng, E., and Song, D. (2020). Acute in vivo testing of a polymer cuff electrode with integrated microfluidic channels for stimulation, recording, and drug delivery on rat sciatic nerve. *J. Neurosci. Methods* 336, 108634.

Fee, M.S., Kozhevnikov, A.A., and Hahnloser, R.H.R. (2004). Neural Mechanisms of Vocal Sequence Generation in the Songbird. *Ann. N. Y. Acad. Sci.* 1016, 153–170.

González-González, M.A., Kanneganti, A., Joshi-Imre, A., Hernandez-Reynoso, A.G., Bendale, G., Modi, R., Ecker, M., Khurram, A., Cogan, S.F., Voit, W.E., et al. (2018). Thin Film Multi-Electrode Softening Cuffs for Selective Neuromodulation. *Sci. Rep.* 8, 16390.

Grill, W.M., and Mortimer, J.T. (2000). Neural and connective tissue response to long-term implantation of multiple contact nerve cuff electrodes. *J. Biomed. Mater. Res.* 50, 215–226.

Lee, Y.J., Kim, H.-J., Do, S.H., Kang, J.Y., and Lee, S.H. (2016). Characterization of nerve-cuff electrode interface for biocompatible and chronic stimulating application. *Sens. Actuators B Chem.* 237, 924–934.

Leonardo, A., and Fee, M.S. (2005). Ensemble coding of vocal control in birdsong. *J. Neurosci.* 25, 652.

McCasland, J.S. (1987). Neuronal control of bird song production. *J. Neurosci.* 7, 23–39.

- Musick, K.M., Rigosa, J., Narasimhan, S., Wurth, S., Capogrosso, M., Chew, D.J., Fawcett, J.W., Micera, S., and Lacour, S.P. (2015). Chronic multichannel neural recordings from soft regenerative microchannel electrodes during gait. *Sci. Rep.* 5, 14363.
- Ölveczky, B.P., Otchy, T.M., Goldberg, J.H., Aronov, D., and Fee, M.S. (2011). Changes in the neural control of a complex motor sequence during learning. *J. Neurophysiol.* 106, 386–397.
- Onuki, Y., Bhardwaj, U., Papadimitrakopoulos, F., and Burgess, D.J. (2008). A Review of the Biocompatibility of Implantable Devices: Current Challenges to Overcome Foreign Body Response. *J. Diabetes Sci. Technol.* 2, 1003–1015.
- Salatino, J.W., Ludwig, K.A., Kozai, T.D.Y., and Purcell, E.K. (2017). Glial responses to implanted electrodes in the brain. *Nat. Biomed. Eng.* 1, 862–877.
- Seo, D., Neely, R.M., Shen, K., Singhal, U., Alon, E., Rabaey, J.M., Carmena, J.M., and Maharbiz, M.M. (2016). Wireless Recording in the Peripheral Nervous System with Ultrasonic Neural Dust. *Neuron* 91, 529–539.
- Struijk, J.J. (1997). The extracellular potential of a myelinated nerve fiber in an unbounded medium and in nerve cuff models. *Biophys. J.* 72, 2457–2469.
- Su, C.-K., Cheng, Y.-W., and Lin, S. (2007). Biophysical and histological determinants underlying natural firing behaviors of splanchnic sympathetic preganglionic neurons in neonatal rats. *Neuroscience* 150, 926–937.
- Ward, W.K. (2008). A Review of the Foreign-body Response to Subcutaneously-implanted Devices: The Role of Macrophages and Cytokines in Biofouling and Fibrosis. *J. Diabetes Sci. Technol.* 2, 768–777.
- Yu, Y., and de Groat, W.C. (2008). Sensitization of pelvic afferent nerves in the *in vitro* rat urinary bladder-pelvic nerve preparation by purinergic agonists and cyclophosphamide pretreatment. *Am. J. Physiol.-Ren. Physiol.* 294, F1146–F1156.